# IMPLICIT NEURAL REPRESENTATION GENERATION WITH HYPERNETWORKS

## ABSTRACT

Implicit Neural Representations (INRs) serve as practical and versatile tools for encoding complex signals through neural networks. While researchers have employed hypernetworks to improve INR adaptability, existing approaches either neglect inter-layer dependencies or lack scalability. We present a novel framework addressing both limitations by regarding INR parameter generation as an optimization process and using the chain rule to capture layer dependencies. We develop a simple yet effective tokenization mechanism enabling Transformer-based hypernetworks to ensure scalability. In addition, we introduce a practical weight initialization strategy that stabilizes training. We conduct extensive experiments across diverse datasets, including 2D images, 3D geometry, and radiance fields. The results consistently demonstrate our method's superiority to state-of-the-art INR generation approaches across all tested datasets.

## 1 INTRODUCTION

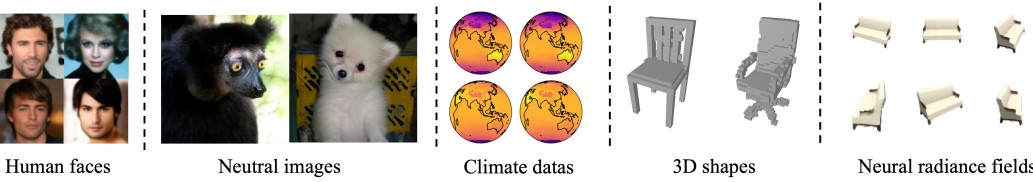

Human faces     Neutral images     Climate datas     3D shapes     Neural radiance fields

Figure 1: Our hypernetwork approach can be widely used in diverse domains, ranging from 2D (e.g., images) to 3D (e.g., occupancy grids and radiance fields) implicit neural representations.

Implicit Neural Representations (INRs) have emerged as an effective paradigm for encoding continuous complex signals from 2D natural images and scientific climate data to 3D geometric shapes and radiance fields by mapping input coordinates directly to output values via neural networks (Sitzmann et al., 2020; Mildenhall et al., 2020). Unlike discrete representations that rely on fixed grids (e.g., pixels for images), INRs offer inherent continuity, compactness, and adaptability, enabling high-fidelity tasks such as photorealistic novel view synthesis.

To enhance the adaptability of INRs researchers have turned to hypernetworks (Ha et al., 2017): specialized networks that generate the weights of target INR models. Hypernetworks enable efficient parameter sharing across tasks while producing task-specific INRs, making them ideal for meta-learning (Chen & Wang, 2022), audio signal processing (Szatkowski et al., 2023), and few-shot learning (Sendera et al., 2023). Recent work has further pushed this direction by treating INR weight generation as a learnable process: diffusion models (Wang et al., 2024) and dataset-conditioned methods (Soro et al., 2024) generate high-performance INR parameters, while Hyper-INR (Wu, 2023) leverages knowledge distillation to speed up training.

Despite recent advances, two critical limitations persist in state-of-the-art hypernetwork-based INR generation: the neglect of inter-layer dependencies and poor scalability. Most hypernetworks generate INR weights independently, ignoring dependencies between consecutive layers. This approach

disrupts the neural network's natural gradient flow, reducing representation quality. Although Hy-PoGen (Ren et al., 2025) attempts to model these dependencies using backpropagation's chain rule, its use of separate MLPs for Jacobian matrices limits scalability when handling large datasets or high-dimensional signals. While recent Transformer-based alternatives (e.g., LDMI (Peis et al., 2025)) improve scalability, they fail to model inter-layer dependencies, which hampers their performance on complex 3D tasks.

To address both limitations, we present a novel hypernetwork framework for INR generation that unifies chain rule-based layer dependency modeling with Transformer-based scalability. Our core insight is that the element-wise dependency structures inherent in Jacobian computations (critical for capturing layer interactions) are precisely the relationships that attention mechanisms excel at modeling - far more effectively than generic MLPs. We introduce three key innovations:

- **Attention-Based Gradient Estimation**. We use cross-attention to model both the forward pass (layer activations) and backward pass (Jacobian matrices) of the target INR, naturally capturing inter-layer dependencies via the chain rule.
- **Tokenization Mechanism**. We tokenize INR parameters (as rows of weight matrices) and hidden states (as individual neurons) to enable uniform processing within the Transformer architecture, ensuring scalability across large datasets and high-dimensional signals.
- **Practical Initialization Strategy**. We design a weight initialization method to stabilize training with SIREN (Sitzmann et al., 2020), a critical requirement for hypernetworks generating complex INR parameters.

As shown in Figure 1, our framework can be widely adapted to different modalities. We conduct extensive experiments to validate our approach across diverse domains: 2D data (CelebA-HQ (Liu et al., 2015) human faces, ImageNet (Russakovsky et al., 2015) natural images, ERA5 (Hersbach et al., 2019) climate data) and 3D data (ShapeNet (Chang et al., 2015) Chairs geometry, neural radiance fields for novel view synthesis (Ramirez et al., 2024)). Our method consistently outperforms state-of-the-art baselines including HyPoGen (Ren et al., 2025) (chain rule-based MLPs) and LDMI (Peis et al., 2025) (Transformer-based without dependency modeling) across all datasets, demonstrating superior reconstruction quality and geometric accuracy.

## 2 RELATED WORKS

### 2.1 IMPLICIT NEURAL REPRESENTATIONS

Implicit Neural Representations (INRs) have emerged as a powerful paradigm for representing continuous signals using neural networks. SIREN (Sitzmann et al., 2020) introduced periodic activation functions, demonstrating that sinusoidal activations enable networks to represent complex natural signals with fine detail. Concurrently, NeRF (Mildenhall et al., 2020) revolutionized 3D scene representation by using coordinate-based neural networks to encode radiance fields, achieving photorealistic novel view synthesis.

Building on these foundational works, (Tancik et al., 2020) showed that Fourier feature mappings enable networks to learn high-frequency functions in low-dimensional domains, addressing the spectral bias inherent in standard neural networks. Instant-NGP (Müller et al., 2022) achieved significant speedups through multiresolution hash encodings, making real-time rendering feasible.

Recent advances have focused on improving training efficiency and representation quality. FR-INR (Shi et al., 2024) introduces Fourier reparameterized training to enhance convergence, while H-SIREN (Gao & Zhang, 2024) explores hyperbolic periodic functions for improved implicit representations. Recent innovations in the field include Quantum INR (Zhao et al., 2024), which bridges quantum computing with neural representations, and Mixture of Experts (Ben-Shabat et al., 2024), which employs specialized expert networks to handle complex signals more effectively.

### 2.2 HYPERNETWORKS

Hypernetworks (Ha et al., 2017) generate the weights of target networks using separate neural networks, enabling efficient parameter sharing and meta-learning. This foundational work has inspired numerous applications across different domains.

HyperSound (Szatkowski et al., 2023) uses hypernetworks for efficient implicit neural audio signal representations. Significant architectural innovations have emerged in the field. HyperFormer (Karimi Mahabadi et al., 2021) introduces parameter-efficient multi-task fine-tuning for Transformers using shared hypernetworks. HyperShot (Sendera et al., 2023) applies kernel hypernetworks to few-shot learning scenarios. MotherNet (Müller et al., 2023) proposes a foundational hypernetwork architecture designed specifically for tabular classification tasks. HyperMAML (Przewięźlikowski et al., 2024) enhances few-shot adaptation by combining hypernetworks with model-agnostic meta-learning. HyPoGen (Ren et al., 2025) introduces hypernetworks to policy generation, enabling generalizable robot manipulation. However, its MLP-based architecture limits its ability to handle large-scale vision datasets.

For novel view synthesis, numerous works have leveraged hypernetworks to achieve impressive results, employing various innovative techniques. Trans-INR (Chen & Wang, 2022) demonstrates the use of Transformers as meta-learners for implicit neural representations, where attention mechanisms are employed to generate INR parameters for novel view synthesis. Similarly, GINR-IPC (Kim et al., 2022) introduced instance pattern composers, enhancing the generalizability of implicit representations for diverse novel view synthesis tasks. GNF-PONP (Gu et al., 2023) proposed partially observed neural processes, addressing challenges in generalizing across novel views. La-GINR (Lee et al., 2023) introduced a locality-aware method, improving the generalizability of implicit neural representations by considering spatial locality, which plays a key role in synthesizing novel views in complex environments. Finally, FM-NNW (Gu & Yeung-Levy, 2025) emphasized how foundation models can enhance hypernetwork architectures, significantly improving their ability to generate high-quality, versatile representations for novel view synthesis tasks.

Recently, hypernetworks have also been introduced for generating INRs. VaMoH (Koyuncu et al., 2023) employs INR-based Variational Autoencoders (VAEs) with hypernetworks that can learn distributions over functions. LDMI (Peis et al., 2025) proposes Transformer-based hypernetworks with latent diffusion models to enhance scalability. Other works, such as HyperDiffusion (Erkoç et al., 2023), achieve 3D and 4D shapes generation. Despite promising results, the lack of inter-layer dependencies in target networks limits their performance.

## 3 PRELIMINARY

**Implicit Neural Representations (INRs).** INRs are neural networks that map input coordinates directly to output values, typically implemented as an MLP $f_\theta : \mathbb{R}^d \to \mathbb{R}^m$ with $L$ layers. For images, an INR learns $f_\theta(x, y) = (r, g, b)$ maps pixel coordinates to colors. In Neural Radiance Fields, $f_\theta$ takes 3D position and viewing direction $(\mathbf{x}, \mathbf{d})$ and outputs color and density $(\mathbf{c}, \sigma)$.

Following the MLP implementation, we denote INRs as a composition of layer functions:

$$f_\theta(\mathbf{x}) = f_{\theta_L} \circ \cdots \circ f_{\theta_1}(\mathbf{x}), \tag{1}$$

where $f_{\theta_i} : \mathbb{R}^{n_{i-1}} \to \mathbb{R}^{n_i}$ represents layer function of layer $i$. Also, hidden states $\mathbf{h}_i = f_{\theta_i}(\mathbf{h}_{i-1}) = \sigma(a_{\theta_i}(\mathbf{h}_{i-1}))$ for $i = 1, \ldots, L-1$, where $a_{\theta_i} : \mathbb{R}^{n_{i-1}} \to \mathbb{R}^{n_i}$ is an *affine* map parameterized by $\theta_i$. The final layer is linear, $f_{\theta_L}(\mathbf{h}_{L-1}) = a_{\theta_L}(\mathbf{h}_{L-1})$. Here $\sigma(\cdot)$ acts element-wise (e.g., ReLU, SIREN, Gaussian), and the full parameter set is $\theta = \{\theta_i\}_{i=1}^L$.

**Hypernetworks for INR Weight Generation.** A hypernetwork $\mathcal{H}_\Omega$ generates the weights $\theta$ for an INR $f_\theta$, where $\Omega$ are the hypernetwork parameters. Given a conditioning input $\mathbf{z}$, the hypernetwork produces $\theta = \mathcal{H}_\Omega(\mathbf{z})$ for the target INR. This enables efficient representation of multiple signals by sharing the hypernetwork $\mathcal{H}_\Omega$ across tasks while generating task-specific INR weights.

There are two major paradigms for INR weight generation, distinguished by supervision approaches:

*Sample-Supervised Generation:* The hypernetwork is trained under the supervision of reconstructing target signals from coordinate-value pairs. The hypernetwork is trained end-to-end by minimizing the reconstruction loss:

$$\mathcal{L}(\Omega) = \sum_{\tau=1}^M \ell\left(f_{\mathcal{H}_\Omega(\mathbf{z}_\tau)}(\mathbf{x}_\tau), \mathbf{y}_\tau\right), \tag{2}$$

where $\ell(\cdot, \cdot)$ is a loss function (e.g., MSE for regression tasks) and $\tau$ represents tasks.

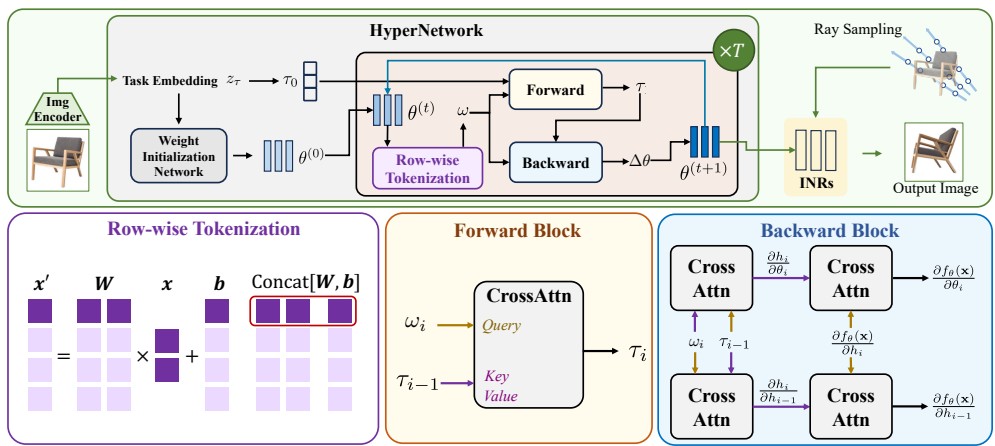

Figure 2: Overview of our proposed framework. The framework begins with network parameters to be optimized that are transformed via row-wise tokenization into structured sequence representations. The forward module employs cross-attention to model the functional dependency $h_i = f_{\theta_i}(h_{i-1})$. The backward module introduces additional cross-attention layers to approximate Jacobians $\partial h_i/\partial h_{i-1}$ and $\partial h_i/\partial \theta_i$, capturing gradient flow relationships essential for parameter generation. Together, this unified attention-based design integrates forward computation and backward differentiation in a principled manner.

*Weight-Supervised Generation:* The hypernetwork is trained directly under the supervision of target INR weights. The hypernetwork is trained to minimize the parameter prediction loss:

$$\mathcal{L}(\Omega) = \sum_{\tau=1}^{M} \|\mathcal{H}_\Omega(\mathbf{z}_\tau) - \theta_\tau^*\|^2, \tag{3}$$

where $\theta_\tau^*$ are the optimal INR parameters from individually trained INRs on the respective tasks.

**Optimization-biased Hypernetwork.** Instead of directly predicting target parameters $\theta^*$ from task embedding $\mathbf{z}_\tau$, recent work HyPoGen (Ren et al., 2025) introduced an optimization-biased hypernetwork that generates parameter updates $\Delta\theta = \mathcal{H}(\mathbf{z}_\tau, \theta)$ and refines them iteratively: $\theta^{(t)} = \lambda\Delta\theta^{(t-1)} + \theta^{(t-1)}$, where $\lambda$ is the step size. The key idea is that parameter generation should mimic the formulation of neural network optimization - using the chain rule of backpropagation:

$$\frac{\partial f_\theta(\mathbf{x})}{\partial \theta_i} = \left( \prod_{j=i+1}^{L} \frac{\partial \mathbf{h}_j}{\partial \mathbf{h}_{j-1}} \right) \frac{\partial \mathbf{h}_i}{\partial \theta_i}, \tag{4}$$

where gradients for parameters at layer $i$ inherently depend on signals from all subsequent layers $j > i$, establishing a principled inter-layer dependency for hypernetwork design.

Optimization-biased hypernetworks improve weight generation by directly incorporating gradient flow structure into parameter synthesis. Accurate gradient computation is essential for effective parameter optimization, with Jacobian matrices being key to computing these gradients through the chain rule. This makes designing better architectural structures for Jacobian estimation critical.

## 4 METHOD

HyPoGen (Ren et al., 2025) treats matrix-valued dependency prediction as an unstructured mapping problem, overlooking the Jacobians' intrinsic role in representing element-wise input–output relationships. We observe that Jacobians naturally encode fine-grained dependencies between inputs and outputs - precisely the pairwise relationships that attention mechanisms excel at capturing. Based on this insight, we propose using cross-attention architectures for modeling neural gradient computation. This approach leverages the attention matrix's natural ability to capture comprehensive dependency patterns and explicit pairwise relationships, rather than relying on MLPs as generic

function approximators that must implicitly learn these structured interactions. Figure 2 presents an overview of our method, which is elaborated in the following sections.

## 4.1 ATTENTION-BASED GRADIENT ESTIMATION

Our approach leverages cross-attention mechanisms to model Jacobian relationships through three key steps. *First*, we tokenize both parameters $\theta_i$ and hidden states $\mathbf{h}_i$ into sequence representations, enabling uniform processing within the attention framework. *Second*, we model the forward pass computation $\mathbf{h}_i = f_{\theta_i}(\mathbf{h}_{i-1})$ using cross-attention, allowing the model to learn how activations at the current layer depend on both the preceding layer's outputs and the current layer's parameters. *Finally*, we employ additional cross-attention modules to approximate the backward pass Jacobians $\frac{\partial \mathbf{h}_i}{\partial \mathbf{h}_{i-1}}$ and $\frac{\partial \mathbf{h}_i}{\partial \theta_i}$, capturing the gradient flow relationships essential for effective parameter generation.

This unified attention-based framework naturally models the element-wise dependency structures inherent in both forward computation and gradient backpropagation, providing a more principled alternative to the generic MLPs used in previous hypernetwork approaches.

### 4.1.1 TOKENIZATIONS

We represent both parameters $\theta_i$ and hidden states $\mathbf{h}_{i-1}$ as sequences of tokens for uniform processing within the attention framework. For the parameter matrix $\theta_i$, we create a row-wise tokenized representation $\boldsymbol{\omega}_i = \{\omega_1, \omega_2, \ldots, \omega_{n_i}\}$ where each token $\omega_j$ is obtained by encoding the corresponding row of $\theta_i$ through a small MLP encoder. To reconstruct the parameter matrix, we decode each token back to its row representation with a small MLP decoder and stack them. Similarly, we tokenize the hidden states as $\boldsymbol{\tau}_i = \{\tau_1, \tau_2, \ldots, \tau_{n_i}\}$ where each token $\tau_j$ represents an individual activation neuron. This tokenization enables cross-attention mechanisms to model element-wise dependencies between parameters and activations through the learned token representations.

### 4.1.2 FORWARD-PASS MODELING

We model the forward pass computation $\mathbf{h}_i = f_{\theta_i}(\mathbf{h}_{i-1})$ using cross-attention between the tokenized representations. For each layer $i$, we use the parameter tokens $\boldsymbol{\omega}_i$ as queries, while the hidden state tokens $\boldsymbol{\tau}_{i-1}$ from the previous layer serve as both keys and values. The cross-attention mechanism computes how each element of the current hidden state depends on the layer parameters[1]:

$$\boldsymbol{\tau}_i = \text{CrossAttn}(\boldsymbol{\omega}_i, \boldsymbol{\tau}_{i-1}, \boldsymbol{\tau}_{i-1}) = \text{softmax}\left(\frac{\boldsymbol{\omega}_i \boldsymbol{\tau}_{i-1}^T}{\sqrt{d_k}}\right)\boldsymbol{\tau}_{i-1} \tag{5}$$

We denote this cross-attention module as $\mathcal{F}_i$ to model the forward-pass computation at layer $i$. This formulation shares a similar computational structure with standard linear layer computation $\mathbf{h}_i = \theta_i \mathbf{h}_{i-1}$. Both operations produce each output element through weighted combinations of input elements:

$$\text{Linear layer:} \quad h_i^{(j)} = \sum_k \theta_i^{(j,k)} h_{i-1}^{(k)} \tag{6}$$

$$\text{Attention:} \quad \tau_i^{(j)} = \sum_k A_{j,k} \tau_{i-1}^{(k)} \tag{7}$$

where $A_{j,k}$ are the learned attention weights. While the weight computation differs – fixed parameters $\theta_i^{(j,k)}$ versus dynamic attention weights $A_{j,k}$ – both mechanisms capture the fundamental structure of combining multiple inputs to produce each output through learned dependency patterns.

### 4.1.3 BACKWARD-PASS MODELING

Following the forward pass simulation, we model the backward pass using cross-attention to approximate both Jacobian computation and chain rule matrix multiplications, adhering to the modality

---

[1]Here we ignore the linear transformations $W_Q, W_K, W_V$ for simplicity and directly use $\boldsymbol{\omega}_i$ as queries and $\boldsymbol{\tau}_{i-1}$ as both keys and values.

consistency principle. The backward pass determines how gradient information flows through the generated INR parameters, directly affecting parameter generation quality.

The core challenge is accurately estimating Jacobian matrices that capture sensitivity relationships between network components. Cross-attention provides a principled framework for learning these dependency patterns, with query-key-value assignments that respect the mathematical structure established in our analogy to differential calculus.

**Intuition for Cross-Attention Formulation:** A key principle guides our cross-attention design for gradient approximation: in cross-attention, keys and values originate from one source while queries come from another, placing them in fundamentally different representational spaces. Since attention computes a weighted linear combination of values, the output necessarily resides in the value space. This modality consistency principle dictates our formulation choices. When approximating a Jacobian $\frac{\partial \mathbf{h}_i}{\partial \mathbf{h}_{i-1}}$ for $\mathbf{h}_i = f(\theta_i, \mathbf{h}_{i-1})$, we must assign $\mathbf{h}_{i-1}$ to both keys and values to ensure the output gradient aligns with $\mathbf{h}_{i-1}$'s representational space. The query then captures the functional dependency—since the Jacobian depends on both $\theta_i$ and $\mathbf{h}_{i-1}$, we use $\theta_i$ as queries to model how parameters influence the sensitivity relationship. This leads naturally to our formulation: $\frac{\partial \mathbf{h}_i}{\partial \mathbf{h}_{i-1}} \approx \text{CrossAttn}(\theta_i, \mathbf{h}_{i-1}, \mathbf{h}_{i-1})$, where $\theta_i$ queries determine attention weights over $\mathbf{h}_{i-1}$ components, and the linear combination of $\mathbf{h}_{i-1}$ values produces gradients in the correct space. This principle ensures representational consistency throughout our backward pass approximation.

**Parameter-to-Activation Jacobians:** The Jacobian $\frac{\partial \mathbf{h}_i}{\partial \theta_i}$ measures how each activation in layer $i$ responds to parameter changes in layer $i$. Following our established formulation where output states determine attention over input components:

$$J_{\theta_i} = \frac{\partial \mathbf{h}_i}{\partial \theta_i} = \text{CrossAttn}\left(\boldsymbol{\tau}_{i-1}, \boldsymbol{\omega}_i, \boldsymbol{\omega}_i\right) \tag{8}$$

where layer activations $\boldsymbol{\tau}_{i-1}$ serve as queries to determine attention weights over parameters $\boldsymbol{\omega}_i$, with parameters as values ensuring gradient modality consistency.

**Hidden-to-Hidden Jacobians:** The inter-layer Jacobian $\frac{\partial \mathbf{h}_{i+1}}{\partial \mathbf{h}_i}$ captures activation propagation between consecutive layers:

$$J_{h_i} = \frac{\partial \mathbf{h}_i}{\partial \mathbf{h}_{i-1}} = \text{CrossAttn}\left(\boldsymbol{\omega}_i, \boldsymbol{\tau}_{i-1}, \boldsymbol{\tau}_{i-1}\right) \tag{9}$$

where parameters $\boldsymbol{\omega}_i$ determine attention over downstream activations $\boldsymbol{\tau}_{i-1}$, approximating local sensitivity relationships through learnable attention patterns.

**Chain Rule Matrix Multiplications via Cross-Attention.** Chain rule computations combine upstream gradients with estimated Jacobians through cross-attention's weighted aggregation mechanism, maintaining numerical stability while capturing essential dependency structures.

**Hidden State Gradients:** Hidden gradients propagate upstream sensitivity through inter-layer Jacobians:

$$\nabla_{h_i} = \frac{\partial f_\theta(\mathbf{x})}{\partial \mathbf{h}_{i-1}} = \text{CrossAttn}\left(\frac{\partial f_\theta(\mathbf{x})}{\partial \mathbf{h}_i}, \frac{\partial \mathbf{h}_i}{\partial \mathbf{h}_{i-1}}, \frac{\partial \mathbf{h}_i}{\partial \mathbf{h}_{i-1}}\right) \approx \frac{\partial f_\theta(\mathbf{x})}{\partial \mathbf{h}_i} \frac{\partial \mathbf{h}_i}{\partial \mathbf{h}_{i-1}} \tag{10}$$

**Parameter Gradients:** Parameter gradients combine upstream gradients with parameter-activation Jacobians:

$$\nabla_{\theta_i} = \frac{\partial f_\theta(\mathbf{x})}{\partial \theta_i} = \text{CrossAttn}\left(\frac{\partial f_\theta(\mathbf{x})}{\partial \mathbf{h}_i}, \frac{\partial \mathbf{h}_i}{\partial \theta_i}, \frac{\partial \mathbf{h}_i}{\partial \theta_i}\right) \approx \frac{\partial f_\theta(\mathbf{x})}{\partial \mathbf{h}_i} \frac{\partial \mathbf{h}_i}{\partial \theta_i} \tag{11}$$

In both cases, upstream gradients serve as queries identifying relevant Jacobian components, while Jacobians provide both attention structure and modality-consistent values, approximating the exact chain rule matrix multiplications through learnable attention weights.

## 4.2 ADAPTIVE PARAMETER INITIALIZATION

INRs require parameters to be initialized within very specific ranges to enable proper training and convergence. In standard INR training, this is typically achieved through careful initialization schemes such as Kaiming or Xavier initialization. However, in hypernetwork-based parameter

generation, ensuring that the generated parameters $\theta$ fall within the appropriate initialization ranges presents a significant challenge.

To address this issue, we introduce an adaptive parameter initialization mechanism that ensures the hypernetwork-generated parameters are aligned with the desired initialization distribution for stable training. Specifically, the scaling factor $s_\tau$ is applied to the hypernetwork output, $\mathcal{H}_\Omega(\mathbf{z}_\tau)_i$, to adjust the variance of the generated weights. The scaling factor $s_\tau(i)$ is computed by taking the ratio of the target variance, $\mathrm{var}_{\mathrm{target}}(i)$, to the actual variance of the hypernetwork output, $\mathrm{var}(\mathcal{H}_\Omega(\mathbf{z}_\tau)_i)$, as follows:

$$s_\tau(i) = \frac{\sqrt{\mathrm{var}_{\mathrm{target}}(i)}}{\sqrt{\mathrm{var}(\mathcal{H}_\Omega(\mathbf{z}_\tau)_i)}} \tag{12}$$

This scaling factor is then applied to the output weights, ensuring that the weights are initialized within the desired range:

$$\tilde{\theta}_i^{(0)} = s_\tau(i) \cdot \mathcal{H}_\Omega(\mathbf{z}_\tau)_i \tag{13}$$

The scaling factor $s_\tau(i)$ is dynamically updated during training, allowing the model to adapt to different layers and tasks, ensuring proper initialization throughout the learning process and promoting stability during training.

The target variance $\mathrm{var}_{\mathrm{target}}(i)$ refers to the desired variance of the initial weights for each layer, and it is typically set based on the requirements of the specific type of INR being trained. For example, in the case of SIREN (Sitzmann et al., 2020), the target variance for the initialization weights is chosen such that the network can effectively represent high-frequency signals. In SIREN, the target variance for each layer $i > 1$ can be set as: $\mathrm{var}_{\mathrm{target}}(i) = \frac{2}{\mathrm{fan\_in}(i)}$ where $\mathrm{fan\_in}(i)$ is the number of input neurons to layer $i$, which helps control the initialization scale to prevent issues such as exploding or vanishing gradients. This target variance ensures that the network can start with weights that are well-suited for learning complex representations, especially for functions with high-frequency components, which are characteristic of INRs.

For more details, please refer to Appendix A.2.2.

## 5 EXPERIMENTS

We employ our hypernetwork approach to generate implicit neural representations (INRs) across multiple data domains. To demonstrate its versatility, we evaluate the method on both 2D and 3D datasets. For 2D data, we use natural images from CelebA-HQ (Liu et al., 2015) and ImageNet (Russakovsky et al., 2015), as well as climate data from ERA5 (Hersbach et al., 2019). For 3D data, we perform geometry reconstruction using ShapeNet Chairs (Chang et al., 2015) and novel view synthesis with pre-trained NeRF datasets (Ramirez et al., 2024). For further details on the loss function and results, see Appendix A.4 and A.5.

### 5.1 2D INR GENERATION

We begin by evaluating our method on 2D implicit neural representation (INR) generation. The experiments in this section all use the autoencoder framework with INR-based decoders. In this setup, an encoder converts the input image into a latent feature, which hypernetworks use to predict the parameters of the decoder (target network). The decoder then reconstructs the color values at each pixel when given the image pixel coordinates as input. To measure reconstruction quality, we use Peak Signal-to-Noise Ratio (PSNR) as our evaluation metric. The numerical results are presented in Table 6, with visual comparisons shown in Figure 3. Details are provided below.

Table 1: 2D reconstruction results measured by PSNR on different datasets.

| Dataset | VAMoH | HyPoGen | LDMI | Ours |
|---|---|---|---|---|
| CelebA-HQ | 23.2 | 16.2 | 24.8 | **27.7** |
| ImageNet | 19.6 | 13.2 | 20.7 | **22.9** |
| ERA5 | 39.0 | 40.0 | 44.6 | **49.3** |

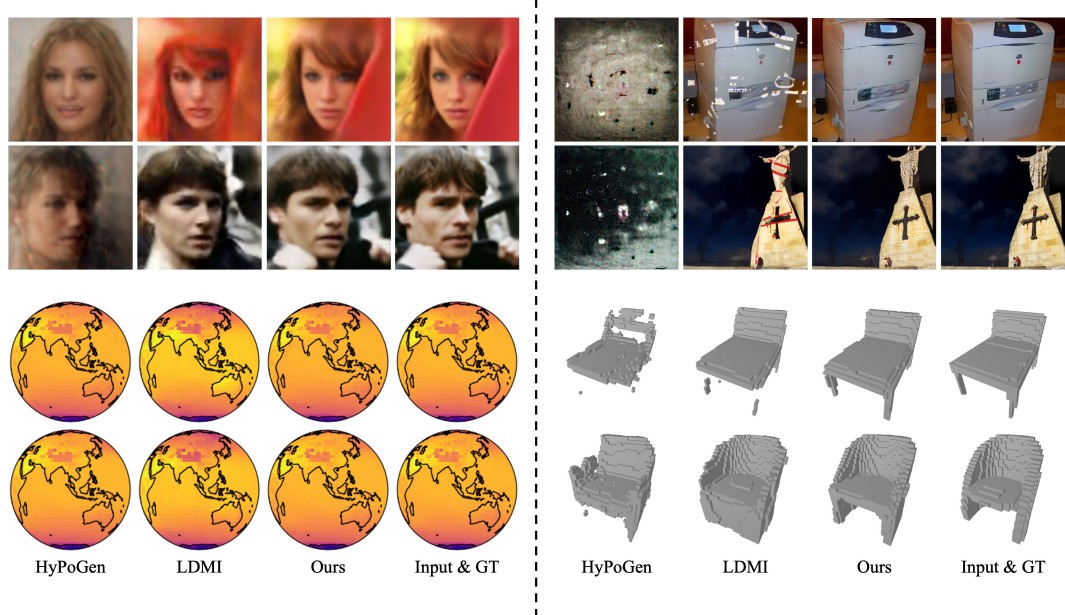

Figure 3: Visual comparison of different 2D INR generation approaches on diverse datasets. Samples from CelebA-HQ are shown in the top left, samples from ImageNet in the top right, samples from ERA5 in the bottom left, and samples from ShapeNet Chairs in the bottom right.

**CelebA-HQ** (Liu et al., 2015). This dataset contains 30,000 human face images from the CelebA dataset, providing a benchmark for face reconstruction tasks using Variational AutoEncoder (VAE). The dataset includes 28,000 training images and 2,000 test images. Following previous works (Dupont et al., 2022; Peis et al., 2025), we use $64 \times 64$ resolution and implement a ResNet-based encoder. All compared methods share the VAE architecture. The low PSNR of HyPoGen (Ren et al., 2025) demonstrates that, despite leveraging layer dependency in the target network, the limited scalability of MLP-based hypernetworks restricts its performance. While LDMI (Peis et al., 2025) employs a Transformer-based hypernetwork, its lack of explicitly modeling layer dependency in the target network limits its effectiveness. Thanks to our chain rule-based formulation with Transformer architecture, our approach achieves the best performance of 26.1 dB PSNR.

**ImageNet** (Russakovsky et al., 2015). For our natural image reconstruction benchmark, we use the standard ILSVRC 2012 subset, which contains 1,431,167 total images across 1,000 classes, comprising 1,281,167 training images, 50,000 validation images, and 100,000 test images. Our method is agnostic to the VAE implementation, so we employ a VQ-VAE (Van Den Oord et al., 2017) based autoencoder architecture for this benchmark. Following LDMI (Peis et al., 2025), we use $256 \times 256$ resolution, and the VQ-VAE is pre-trained and shared across all compared methods. Despite using a pre-trained VQ-VAE, HyPoGen (Ren et al., 2025) failed to converge during training, resulting in messy reconstructions. This confirms that MLP-based hypernetworks have limited capacity when handling large-scale datasets with diverse samples. Our method achieves a PSNR value of 22.9 dB, outperforming LDMI (Peis et al., 2025). As shown in Figure 3, LDMI's results contain noticeable artifacts or floating points, while our reconstructions more closely match the ground truth. These results further verify the effectiveness of our approach that leverages layer dependencies with the chain rule based formulation.

**ERA5** (Hersbach et al., 2019). To validate the broader applicability of our approach, we test our method on ERA5 climate data. This dataset features hourly atmospheric measurements on a 31km grid from ECMWF's fifth-generation global climate reanalysis. The polar climate data represents

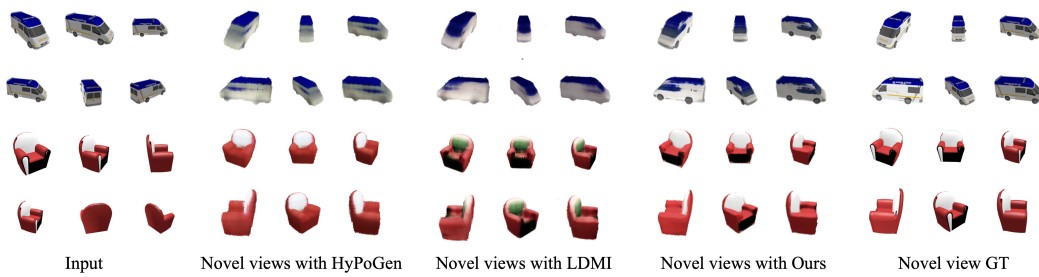

Input    Novel views with HyPoGen    Novel views with LDMI    Novel views with Ours    Novel view GT

Figure 4: Visual comparison of different 3D INR generation approaches on the NeRF dataset.

a unique scientific domain that challenges our method's ability to handle complex spatial-temporal patterns. Specifically, the data is stored in matrices similar to 2D images. We split the dataset into 8,510 training samples, 1,126 validation samples, and 2,420 test samples. We use a resolution of $46 \times 90$, where the value of each pixel represents the temperature of the corresponding region. We apply a standard VAE architecture for all the compared methods. We are pleased to observe that our approach achieves the best results among all compared methods. This verifies both the generalization ability and effectiveness of our methodology.

## 5.2    3D INR GENERATION

In our evaluation of 3D implicit neural representation (INR) generation, we use two datasets for 3D shape reconstruction and novel view synthesis.

**ShapeNet Chairs** (Chang et al., 2015). This dataset contains 6,778 3D chair model instances with aligned object poses and is widely used in 3D shape reconstruction tasks. We split

Table 2: Results on ShapeNet Chairs.

| Method | VAMoH | HyPoGen | LDMI | Ours |
|---|---|---|---|---|
| Accuracy | 96.8 % | 95.2 % | 97.3% | **97.9%** |

the dataset into 5,422 training samples, 677 validation samples, and 679 test samples. Following LDMI (Peis et al., 2025), we convert each object instance into occupancy grids with $32 \times 32 \times 32$ resolution. The VAE employs 3D-convolutional networks. Table 13 shows our numerical results on shape reconstruction. We evaluate geometry accuracy by calculating the percentage of correct occupancy predictions. Our method outperforms other approaches in comparison. As Figure 3 shows, HyPoGen (Ren et al., 2025) typically reconstructs only the main body of objects, omitting details like chair legs. LDMI similarly produces incomplete reconstructions with wrong geometry and rough surfaces. Our method, however, generates complete reconstructions with smoother surfaces.

**Pre-trained NeRF datasets** (Ramirez et al., 2024). The dataset includes 103K objects divided into 98,625 training objects and 3,961 test objects. Each object contains 36 rendered images with their corresponding camera parameters. Unlike other datasets with VAE problem settings that focus on reconstructing input images, this dataset emphasizes novel view synthesis. It uses condition images and their camera parameters as input to learn a Neural Radiance Field (NeRF) for each object, which then renders new images from novel viewpoints. During training, we randomly select 6 images as condition views and sample another 6 novel views to be synthesized.

Table 3 compares the performance of our approach against several baselines under three different supervision methods: Weight-only supervision, where only the pre-trained NeRF weights are used for supervision. Image-only supervision, which uses the ground truth images from novel views as supervision. Weight & Image supervision, which combines both the pre-trained NeRF weights and the ground truth images from novel views as supervision.

Our experimental results show that our approach outperforms all other methods across these supervision settings. With image supervision, we observe noticeable improvements in rendering quality across most methods. In Figure 4, we show the synthesized novel view images under the Weight & Image supervision setting. We can observe that our generations better align with ground truth

images. This consistently validates the effectiveness of our approach in generating high-quality 3D INRs.

Table 3: Novel view synthesis on NeRF dataset.

| Supervision | HyperDiffusion | HyPoGen | LDMI | Ours |
|---|---|---|---|---|
| Weight only | 20.0 | 19.9 | 19.3 | **22.0** |
| Image only | N/A | 24.9 | 26.2 | **28.8** |
| Weight & Image | 18.1 | 25.5 | 26.5 | **27.8** |

## 5.3 ABLATION STUDIES

We conduct ablation studies to validate the key components of our framework. All experiments use consistent training settings to isolate individual contributions. The results are reported in Table 4.

**Reversed Q-KV Assignment.** This ablation tests the impact of reversing the query-key-value (Q-KV) assignments in the cross-attention module. In this setup, we swap the roles of the query and key-value pairs during the attention mechanism. The results show a significant performance drop (PSNR = 18.5), which demonstrates the critical role of correctly aligning Q, K, and V elements in the attention mechanism.

**Column-Based Tokenization.** In this experiment, we replace our row-wise tokenization of the weight matrices with column-wise tokenization. This modification results in a noticeable performance degradation (PSNR = 19.8), indicating that row-wise tokenization is key for efficiently capturing inter-layer dependencies. Row-wise tokenization enables more effective scaling and dependency modeling across layers, which is essential for good performance.

**No Token Initialization Network.** We investigate the effect of removing the token initialization mechanism from the hypernetwork. In this case, we do not initialize the target network's weights through the hypernetwork's tokenization procedure. The performance drops to 25.5 PSNR, indicating that the initialization process is crucial for stabilizing training. For the specific **detail** of the **Token Initialization Network**, please refer to Appendix A.2.1.

**Single Layer Optimization ($T = 1$).** This ablation explores the effect of reducing the number of optimization layers ($T$) in the hypernetwork to a single layer. In the full method, the hypernetwork performs iterative weight optimization across 3 layers ($T = 3$), which allows for a more refined adjustment of the target network's weights. The **Single Layer Optimization ($T = 1$)** ablation limits this process to a single layer of optimization, reducing the model's ability to capture deeper dependencies between layers. More methodological details can be found in Appendix A.2.4.

**Full Method (Default).** The final row reports the performance of our full method, which includes all design choices: proper query-key-value assignments, row-wise tokenization, adaptive initialization, and multi-layer optimization. Our method achieves the best performance (PSNR = 27.7), demonstrating the synergistic benefits of all components working together.

These ablation studies confirm that each of the proposed components—correct Q-KV assignments, row-wise tokenization, adaptive initialization, and multi-layer optimization—are crucial for achieving high performance in INR generation.

Table 4: Ablation studies on the NeRF dataset.

| Method | PSNR |
|---|---|
| Reversed Q-KV Assignment | 18.5 |
| Column-Based Tokenization | 19.8 |
| No Token Initialization Network | 25.5 |
| Single Layer Optimization | 27.4 |
| Full Method (Default) | **27.8** |

## 6 Conclusion

This paper presents a novel hypernetwork framework for implicit neural representation generation that unifies chain rule-based dependency modeling with Transformer scalability. Our attention-based gradient estimation naturally captures inter-layer dependencies, while row-wise tokenization and adaptive initialization ensure practical deployment. Extensive experiments across 2D and 3D datasets demonstrate state-of-the-art performance, with particularly strong results on complex datasets requiring accurate gradient flow modeling. By bridging optimization theory and architectural design, our work establishes that explicitly modeling neural network gradient structure leads to superior parameter generation, advancing the field of implicit neural representation learning.

## 7 Ethics Statement

This work strictly adheres to the ICLR Code of Ethics. Our research focuses on developing novel hypernetwork architectures for implicit neural representation generation and does not involve human subjects, animal experiments, or sensitive data collection. All experiments were conducted using publicly available datasets including CelebA-HQ, ImageNet, ERA5, ShapeNet, and pre-trained NeRF datasets, with no privacy concerns or ethical issues arising from data usage.

The proposed methodology advances fundamental research in neural representation learning without direct applications that could cause harm. We acknowledge that improvements in neural network generation capabilities have broader implications for computational efficiency and resource usage. Our approach promotes more efficient parameter generation through principled architectural design, potentially reducing computational requirements compared to existing methods.

All authors have thoroughly reviewed this work and confirm full compliance with ethical guidelines. We believe our contributions advance scientific understanding of hypernetwork architectures and implicit neural representations in a responsible manner that benefits the research community.

## 8 Reproducibility Statement

We have taken comprehensive measures to ensure the reproducibility of our research. Complete architectural details of our hypernetwork framework, including the attention-based gradient estimation mechanism, tokenization procedures, and initialization strategies, are provided in Section 4 and Appendix A.2. All mathematical formulations for forward and backward pass modeling are explicitly defined with equations 5, 6, 7, 8, 9, 10, 11 in the main text.

Experimental configurations are thoroughly documented in Appendix A.3.2, including learning rates, optimizer settings, batch sizes, and model hyperparameters for each dataset (Table 7). Training procedures follow standard protocols with Adam optimizer using $\beta_1 = 0.9$, $\beta_2 = 0.999$, and learning rate $5 \times 10^{-5}$ across all experiments. Target INR architectures are specified for each dataset: 5-layer SIREN networks with 256 hidden units for 2D datasets, 3-layer networks with 128 units for ShapeNet Chairs, and simplified NeRF architectures for novel view synthesis.

Our experiments utilize established public benchmarks with standard evaluation protocols. Dataset splits, resolution specifications, and evaluation metrics are clearly defined in Section 5 and Appendix A.3.1. All baseline implementations follow published configurations to ensure fair comparison.

Upon acceptance, we commit to releasing our complete implementation including trained models, evaluation scripts, and configuration files to facilitate reproduction of all reported results. The codebase will include documentation for replicating experiments across all tested datasets and extending our framework to new domains.

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

# A  APPENDIX

## A.1  THE USE OF LARGE LANGUAGE MODELS

Large language models (LLMs) were employed in this work as auxiliary tools to support the writing process. Specifically, they assisted in refining the clarity of expression, ensuring coherence across sections, and suggesting stylistic adjustments to align with academic writing standards. All substantive ideas, experimental designs, and conclusions, however, remain the intellectual contributions of the authors.

## A.2  MORE DETAILS OF THE METHOD

### A.2.1  TOKEN INITIALIZATION NETWORK

Previous work HyPoGen (Ren et al., 2025) starts with a shared parameter $\theta$ across all tasks and uses chained multiple blocks to iteratively update $\theta$. In our work, instead of using a shared parameter, we use a small transformer network that directly gives an initial guess of the parameter tokens $\omega^{(0)} = \mathcal{P}_{\text{init}}(z_\tau)$, which significantly boosts the convergence speed.

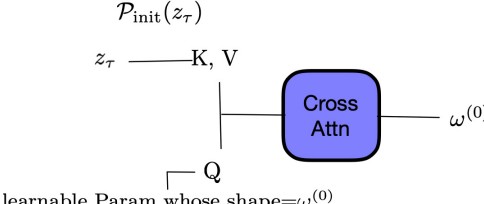

Figure 5: Architecture of the token initialization network $\mathcal{P}_{\text{init}}$. For each layer $i$ of the target INR, learnable parameter queries $\mathcal{Q}_i$ attend to the shared task embedding $z_\tau$ through independent cross-attention modules to generate layer-specific initial parameter tokens $\omega_i^{(0)}$.

The token initialization network $\mathcal{P}_{\text{init}}$ generates task-specific initial parameter tokens for each layer of the target INR through dedicated cross-attention modules. For each layer $i$ with $n_i$ parameters arranged in a row-wise manner, we initialize learnable parameter queries $\mathcal{Q}_i \in \mathbb{R}^{1 \times n_i \times d}$, where $d$ is the embedding dimension. Each layer has its own dedicated cross-attention module consisting of 4 transformer layers with 128-dimensional hidden states to ensure layer-specific parameter generation.

During forward pass, the task embedding $z_\tau$ serves as both keys and values for all cross-attention operations:

$$\omega_i^{(0)} = \text{CrossAttn}_i(\mathcal{Q}_i, z_\tau, z_\tau)$$

where $\mathcal{Q}_i$ is expanded to batch dimension $B$. Each cross-attention module independently extracts task-relevant information from $z_\tau$ while the learnable queries $\mathcal{Q}_i$ capture layer-specific initialization patterns. The output $\omega_i^{(0)} \in \mathbb{R}^{B \times n_i \times d}$ provides the initial parameter tokens for layer $i$.

This design ensures that each layer receives initialization tailored to both the specific task (through $z_\tau$) and its structural role in the network (through layer-specific queries and attention modules). By providing personalized starting points rather than shared parameters across tasks, this approach significantly accelerates convergence in the subsequent optimization blocks.

### A.2.2  COMPARISON OF TOKEN INITIALIZATION NETWORK AND ADAPTIVE PARAMETER INITIALIZATION

**Token Initialization Network:**  The **Token Initialization Network** is responsible for generating the **initial weights**, which serve as the input to the hypernetwork. This network generates task-specific tokens for each layer of the target INR model. The initialization procedure involves tokenizing the parameters and inputting them into the hypernetwork for processing. The goal is to

ensure that the input tokens are initialized in a way that leads to effective learning. By structuring and personalizing these initial weights, the Token Initialization Network stabilizes the learning process and accelerates convergence, providing a solid foundation for subsequent training.

**Adaptive Parameter Initialization:** On the other hand, **Adaptive Parameter Initialization** focuses on the initialization of the **hypernetwork output**. Specifically, it ensures that the predicted target network parameters match the required distribution for stable training. The mechanism achieves this by applying a dynamic scaling factor $s_\tau$, which adjusts the output of the hypernetwork to align with the desired initialization statistics. This scaling ensures that the generated parameters are within the optimal range for stable and efficient convergence during training. By correcting any discrepancies between the hypernetwork's natural output distribution and the desired target, Adaptive Parameter Initialization facilitates smoother and faster convergence.

**Ablation Study on Adaptive Parameter Initialization:** We conducted an ablation study on the CelebA-HQ dataset and ERA5 dataset by removing the scaling factor $s_\tau$. As shown in Table 5, on the CelebA-HQ dataset, the PSNR dropped to 11.8, a stark contrast to 27.7 when adaptive initialization was applied. Similarly, the ERA5 dataset also showed a notable performance gap, where the PSNR was 8.6 without adaptive initialization, compared to 49.3 when the scaling factor $s_\tau$ was used. This demonstrates that adaptive initialization is essential for SIREN-based INRs, ensuring the weights are within the proper range for stable convergence and effective learning.

Table 5: Ablation Study on Adaptive Parameter Initialization measured by PSNR.

| Dataset | without adaptive initialization | using adaptive initialization |
|---|---|---|
| CelebA-HQ | 11.8 | **27.7** |
| ERA5 | 8.6 | **49.3** |

### A.2.3 ABLATION STUDY ON DEPENDENCY MODELING

As described in Section 4.1.2 and Section 4.1.3, we use cross-attention to capture how parameters at different layers influence each other during computation. Meanwhile, LDMI employs cross-attention between task embedding and parameter tokens. To validate the importance of parameter inter-dependency modeling, we conducted an ablation study by modifying LDMI's decoder to use self-attention among parameter tokens, similar to our approach. Table 6 presents the results of this enhanced LDMI variant across all datasets. Although the self-attention enhanced LDMI variant showed improvement in all cases (as shown in Table 6), its performance was still limited compared to our proposed approach. This suggests that while self-attention can capture intra-layer dependencies, it is the inter-layer parameter dependencies modeled by cross-attention that play a crucial role in enhancing the overall performance. The results from this ablation study further support our claim that modeling parameter interactions across layers is essential for stable learning and effective generalization.

### A.2.4 LEARNABLE LEARNING RATE

In our model, the learning rate $\lambda$ is learned during the forward pass and is used to update the parameters of target net predicted by the hypernetwork.

The learning rate for each layer $i$ is predicted using a cross-attention mechanism. Specifically, for each layer, we compute the attention between the previous layer's hidden states $\tau_{i-1}$ and the current layer's parameter tokens $\omega_i$. The calculation proceeds as:

$$\lambda_i' = \text{CrossAttn}(\tau_{i-1}, \omega_i, \omega_i) = \text{softmax}\left(\frac{\tau_{i-1}\omega_i^T}{\sqrt{d_k}}\right)\omega_i$$

After obtaining the cross-attention output $\lambda_i'$, it is flattened and passed through a MLP to generate the final learning rate $\lambda_i$ for each layer. The shape of the resulting output is $[B, 1]$, where $B$ is the batch size, representing the scalar learning rate for each layer.

The learned learning rate $\lambda_i$ is then used to scale the predicted parameter update $\Delta\theta_i$ for the target network. The update rule is given by:

$$\theta_i^T = \theta_i^{T-1} + \lambda_i \cdot \Delta\theta_i^{T-1}$$

Where: $\theta_i^T$ represents the updated parameters of layer $i$ after $T$ update iterations, $\theta_i^{T-1}$ represents the parameters of layer $i$ at the previous iteration, $\lambda_i$ is the learned learning rate for layer $i$, $\Delta\theta_i^{T-1}$ is the predicted parameter update for the target network layer $i$ at the previous iteration. We use $T = 3$ in our experiments across all datasets to ensure a balance between model expressiveness and computational efficiency.

Table 6: Performance comparison of dependency modeling approaches across datasets.

| Dataset | LDMI | LDMI with self-attention decoder | Ours |
|---|---|---|---|
| CelebA-HQ PSNR | 24.8 | 25.9 | **27.7** |
| ERA5 PSNR | 44.6 | 45.4 | **49.3** |
| ShapeNet Chairs % | 97.3 | 97.5 | **97.9** |
| Pre-trained NeRF PSNR | 26.5 | 26.6 | **27.8** |

## A.3 MORE DETAILS OF THE EXPERIMENT

### A.3.1 BASELINE METHODS

We compare our approach against four established baselines representing different paradigms for hypernetwork-based neural field generation.

**VaMoH (Variational Mixture of HyperGenerators for Learning Distributions Over Functions)** (Koyuncu et al., 2023) leverages a normalizing flow to define the prior distribution and employs a mixture of hypernetworks to parametrize the data log-likelihood. This architecture enables effective learning of rich distributions over continuous functions while providing strong inference capabilities for tasks such as missing data imputation. VaMoH demonstrates high expressive capability and interpretability across diverse data types including images, voxels, and climate data, making it a strong baseline for function space generative modeling.

**LDMI (Hyper-Transforming Latent Diffusion Models)** (Peis et al., 2025) combines hypernetworks with latent diffusion models to generate implicit neural representations. The method operates in a compressed latent space and employs Transformer-based architectures to decode latent codes into INR parameters. LDMI represents the state-of-the-art in latent-space approaches for neural field synthesis, demonstrating strong performance across multiple domains through its hybrid architecture.

**HyPoGen (HyPoGen: Optimization-Biased Hypernetworks for Generalizable Policy Generation)** (Ren et al., 2025) introduces optimization-biased hypernetworks that generate network parameters by modeling the optimization process. While originally developed for policy generation tasks, HyPoGen's core hypernetwork architecture provides a strong baseline for INR generation. The method's optimization-aware design makes it particularly relevant for our weight-space generation task.

**HyperDiffusion (HyperDiffusion: Generating Implicit Neural Fields with Weight-Space Diffusion)** (Erkoç et al., 2023) operates directly in the weight space of neural networks, using diffusion models to generate MLP parameters for implicit neural fields. Unlike latent-space approaches, HyperDiffusion performs diffusion directly on network weights. For our experiments, we adapt HyperDiffusion with conditional structure by employing DINO encoders (Caron et al., 2021) for NeRF datasets to provide appropriate conditioning signals.

These baselines span the spectrum of current approaches: VaMoH emphasizes variational inference and mixture modeling in function spaces, LDMI focuses on latent-space efficiency, HyPoGen emphasizes optimization bias in hypernetwork design, and HyperDiffusion leverages diffusion processes directly in parameter space.

### A.3.2 IMPLEMENTATION DETAILS

**Training Configuration** All experiments were conducted on NVIDIA H800 GPUs with mixed precision training. We employ the Adam optimizer with $\beta_1 = 0.9$, $\beta_2 = 0.999$, and a fixed learning rate of $5 \times 10^{-5}$ across all datasets without learning rate scheduling.

**Architecture Specifications** Our hypernetwork architecture scales with dataset complexity, as shown in Table 7. The task tokenizer dimension and hidden dimensions are adjusted based on the signal complexity: simpler datasets like ShapeNet Chairs use 64-dimensional representations, while complex datasets like Pre-trained NeRF require 768 dimensions.

Table 7: Model hyperparameter across different datasets.

|  | CelebA-HQ 64×64 | ImageNet | ERA5 | Chairs | Pre-trained NeRF |
|---|---|---|---|---|---|
| Learning rate | 5e-5 | 5e-5 | 5e-5 | 5e-5 | 5e-5 |
| Optimizer | Adam | Adam | Adam | Adam | Adam |
| Mini-batch size | 64 | 128 | 128 | 32 | 96 |
| Task Tokenizer dim | 128 | 256 | 128 | 64 | 768 |
| Each Block Hidden dim | 128 | 768 | 768 | 64 | 768 |
| Transformer layers | 1 | 1 | 1 | 1 | 1 |
| Token len | 384,257,257,257,4 | 384,257,257,257,4 | 342,257,257,257,2 | 171,129,2 | 64,64,64,16 |
| INR type | SIREN | SIREN | SIREN | SIREN | NeRF |
| INR layers | 5 | 5 | 5 | 3 | 3 |
| INR hidden dim | 256 | 256 | 256 | 128 | 64 |
| INR $\omega$ / encode frequency | 30 | 30 | 30 | 30 | 24 |

**Target INR Configurations** For 2D datasets (CelebA-HQ, ImageNet, ERA5), we employ 5-layer SIREN networks with 256 hidden units and frequency parameter $\omega = 30$, following the original SIREN configuration. For 3D geometry (ShapeNet Chairs), we use a more compact 3-layer architecture with 128 hidden units. The Pre-trained NeRF dataset uses a simplified NeRF architecture without view direction dependency: positional encoding with 24 frequencies for spatial coordinates only, followed by 3 fully-connected layers with 64 hidden units without using bias terms.

**Training and Testing Performance** Tables 8 and Table 9 provide comprehensive training and testing performance analysis across all datasets. Table 13 reports training throughput, GPU memory consumption with maximum batch sizes, and model parameters, while Table 14 presents test-time GPU memory usage and inference latency with batch size set to 1.

Table 8: Comparison of Train Time, memory consumption, and model size across datasets.

| Dataset | Throughput (sample/s) | | | GPU memory(GB)/Max Batch Size | | | Params (M) | | |
|---|---|---|---|---|---|---|---|---|---|
|  | HyPoGen | LDMI | Ours | HyPoGen | LDMI | Ours | HyPoGen | LDMI | Ours |
| CelebA-HQ | 630 | 600 | 213 | 73.32 / 900 | 77.71 / 420 | 78.21 / 256 | 128.8 | 40.5 | 41.2 |
| ImageNet | 142 | 52 | 70 | 75.97/60 | 74.35 / 40 | 78.12 / 42 | 578.7 | 544.9 | 124.5 |
| ERA5 | 3111 | 616 | 170 | 77.68 / 1400 | 72.97 / 128 | 78.24 / 252 | 93.1 | 48.9 | 21.3 |
| Shapes3D | 2660 | 1500 | 806 | 74.91 / 700 | 74.79 / 420 | 73.50 / 576 | 8.3 | 30.9 | 4.1 |
| Pretrained NeRF | 15 | 16 | 30 | 76.03 / 56 | 79.02/16 | 75.58 / 40 | 1049.9 | 515.5 | 482.5 |

Table 9: Comparison of test time and memory consumption with batch size set to 1.

| Dataset | Test-time GPU Memory (GB) | | | Inference Time (ms) | | |
|---|---|---|---|---|---|---|
|  | HyPoGen | LDMI | Ours | HyPoGen | LDMI | Ours |
| CelebA-HQ | 1.037 | 0.854 | 0.389 | 5 | 18 | 166 |
| ImageNet | 4.757 | 10.32 | 1.382 | 8 | 27 | 89 |
| ERA5 | 0.769 | 0.285 | 0.238 | 9 | 13 | 81 |
| Shapes3D | 0.172 | 0.138 | 0.142 | 3 | 9 | 55 |
| Pretrained NeRF | 9.286 | 7.43 | 6.08 | 169 | 167 | 205 |

### A.4   LOSS FUNCTION

For the CelebA-HQ, ImageNet, ERA5, and ShapeNet Chairs datasets, we follow the exact training losses used in LDMI. For the Pre-trained NeRF dataset, we employ a simple L1 image loss and/or an L1 weight loss.

**Note: N** is the total number of samples in each mini-batch used for training. $\lambda_{\text{disc}}$ is a hyperparameter that scales the discriminator loss in the total loss function. $\lambda_{\text{kl}}$ is a hyperparameter that scales the KL divergence loss. The following sections will not repeat the explanation of these parameters.

#### A.4.1   CELEBA-HQ

For the generator:

$$\mathcal{L}_{\text{total}} = \mathcal{L}_{\text{rec}} + \lambda_{\text{disc}} \cdot \mathcal{L}_{\text{g}} + \lambda_{\text{kl}} \cdot \mathcal{L}_{\text{kl}}$$

Where:

$$\mathcal{L}_{\text{rec}} = \frac{1}{N} \sum_{i=1}^{N} w_i \cdot \left( \frac{|x_i - \hat{x}_i|}{\exp(\log \sigma^2)} + \log \sigma^2 \right)$$

is the **reconstruction loss**, computed as the L1 loss between the real input images $(x_i)$ and the reconstructed images $(\hat{x}_i)$, adjusted by the variance $\sigma^2$ and weighted by the factor $w_i$.

$$\mathcal{L}_{\text{kl}} = \frac{1}{N} \sum_{i=1}^{N} \text{KL}(q(z_i) \parallel p(z_i))$$

is the **Kullback-Leibler divergence loss**, computed between the posterior $q(z_i)$ and the prior $p(z_i)$ of the latent variable $z_i$, encouraging the posterior to approximate the prior distribution.

$$\mathcal{L}_{\text{g}} = -\frac{1}{N} \sum_{i=1}^{N} \log D(\hat{x}_i)$$

is the **generator's adversarial loss**, where $D(\hat{x}_i)$ represents the discriminator's output for the reconstructed image $\hat{x}_i$, which the generator aims to maximize in order to fool the discriminator into classifying the fake images as real.

For the discriminator:

$$\mathcal{L}_{\text{discriminator}} = \lambda_{\text{disc}} \cdot \mathcal{L}_{\text{d}}$$

Where:

$$\mathcal{L}_{\text{d}} = \frac{1}{N} \sum_{i=1}^{N} \left[ \log D(x_i) + \log(1 - D(\hat{x}_i)) \right]$$

is the **discriminator's loss**, computed as the sum of the log-likelihood for real images $(D(x_i))$ and the log-likelihood for fake images $(D(\hat{x}_i))$. The discriminator attempts to classify real images as real and reconstructed images as fake.

#### A.4.2   IMAGENET

The total loss function is composed of two parts: the **Autoencoder Loss** and the **Discriminator Loss**.

The **Autoencoder Loss** ensures that the model reconstructs the image accurately and regularizes the latent space through codebook utilization. The total **Autoencoder Loss** is defined as:

$$\mathcal{L}_{\text{autoencoder}} = \mathcal{L}_{\text{rec}} + \lambda_{\text{codebook}} \cdot \mathcal{L}_{\text{codebook}}$$

The **reconstruction loss ($\mathcal{L}_{\text{rec}}$)** is the sum of these two components:

$$\mathcal{L}_{\text{rec}} = \mathcal{L}_{\text{pixel}} + \lambda_{\text{perceptual}} \cdot \mathcal{L}_{\text{perceptual}}$$

Where $\lambda_{\text{perceptual}}$ is a hyperparameter that controls the relative weight of the perceptual loss in the total loss.

1. **Pixel-wise Loss** ($\mathcal{L}_{\text{pixel}}$): Measures the difference between the original and reconstructed images at the pixel level. This can be expressed using L1 loss:

$$\mathcal{L}_{\text{pixel}}(x, y) = \|x - y\|_1$$

Where: $x$ is the original image, $y$ is the reconstructed image.

2. **Perceptual Loss**($\mathcal{L}_{\text{perceptual}}$): Measures perceptual similarity between the original and reconstructed images using a feature-based metric such as LPIPS:

$$\mathcal{L}_{\text{perceptual}} = \text{LPIPS}(x, y)$$

The **codebook Loss** ($\mathcal{L}_{\text{codebook}}$) is used in VQ-VAE to regularize the discrete latent space. It ensures that the continuous latent representations map to the nearest discrete code in the codebook. The codebook loss is defined as:

$$\mathcal{L}_{\text{codebook}} = \|\mathbf{z} - \mathbf{e}\|_2^2$$

Where: $\mathbf{z}$ is the continuous latent representation, $\mathbf{e}$ is the nearest code in the codebook.

The **Discriminator Loss** is responsible for training the discriminator to distinguish between real and fake images which is defined as:

$$\mathcal{L}_{\text{discriminator}} = \lambda_{\text{disc}} \cdot \text{Weight}_{\text{adjusted}} \cdot \mathcal{L}_{\text{d}}^{\text{hinge}}$$

1. **Hinge Discriminator Loss ($\mathcal{L}_{\text{d}}^{\text{hinge}}$)**: The **Hinge Discriminator Loss** is defined as:

$$\mathcal{L}_{\text{d}}^{\text{hinge}} = 0.5 \cdot (\mathcal{L}_{\text{real}} + \mathcal{L}_{\text{fake}})$$

Where: $\mathcal{L}_{\text{real}} = \frac{1}{N}\sum_{i=1}^{N} \text{ReLU}(1 - \text{logits\_real}_i)$ is the loss for real images. $\mathcal{L}_{\text{fake}} = \frac{1}{N}\sum_{i=1}^{N} \text{ReLU}(1 + \text{logits\_fake}_i)$ is the loss for fake images.

2. **Adaptive Weighting (Weight$_{\text{adjusted}}$)**: The **Adaptive Weighting** is applied to the discriminator loss to scale its contribution during training. It is defined as:

$$\text{Weight}_{\text{adjusted}} = \begin{cases} 0 & \text{if global\_step} < \text{threshold}, \\ \text{weight} & \text{otherwise.} \end{cases}$$

### A.4.3   ERA5

The total loss function is

$$\mathcal{L}_{\text{total}} = \mathcal{L}_{\text{rec}} + \lambda_{\text{kl}} \cdot \mathcal{L}_{\text{kl}}$$

Where:

$$\mathcal{L}_{\text{rec}} = \frac{1}{N}\sum_{i=1}^{N} \left( \frac{|x_i - \hat{x}_i|}{\exp(\log \sigma^2)} + \log \sigma^2 \right)$$

is the **reconstruction loss**, computed as the L1 loss between the original temperature ($x_i$) and the reconstructed temperature ($\hat{x}_i$), adjusted by the variance $\sigma^2$.

$$\mathcal{L}_{\text{kl}} = \frac{1}{N}\sum_{i=1}^{N} \text{KL}(q(z_i) \parallel p(z_i))$$

is the **Kullback-Leibler divergence loss**, computed between the posterior $q(z_i)$ and the prior $p(z_i)$ of the latent variable $z_i$, encouraging the posterior to approximate the prior distribution.

### A.4.4 SHAPENET

The total loss function is

$$\mathcal{L}_{\text{total}} = \mathcal{L}_{\text{rec}} + \lambda_{\text{kl}} \cdot \mathcal{L}_{\text{kl}}$$

Where:

$$\mathcal{L}_{\text{rec}} = \frac{1}{N} \sum_{i=1}^{N} w_i \cdot (-x_i \log \sigma(\text{logits}_i) - (1 - x_i) \log(1 - \sigma(\text{logits}_i)))$$

is the **binary cross-entropy loss**, where: $x_i$ is the binary input (either 0 or 1), $\sigma(\text{logits}_i)$ is the sigmoid of the logits (predicted probability for the input being 1).

$$\mathcal{L}_{\text{kl}} = \frac{1}{N} \sum_{i=1}^{N} \text{KL}(q(z_i) \parallel p(z_i))$$

is the **Kullback-Leibler divergence** loss, computed between the posterior $q(z_i)$ and the prior $p(z_i)$ of the latent variable $z_i$, encouraging the posterior to approximate the prior distribution.

### A.4.5 PRE-TRAINED NERF

The total loss function is

$$\mathcal{L}_{\text{total}} = \lambda_{\text{weight}} \cdot \mathcal{L}_{\text{weight}} + \lambda_{\text{img}} \cdot \mathcal{L}_{\text{img}}$$

Where:

$$\mathcal{L}_{\text{weight}} = \frac{1}{N} \sum_{i=1}^{N} \|w_i^{\text{pred}} - w_i^{\text{gt}}\|_1$$

is the **weight loss**, computed as the L1 loss between the predicted weights ($w_i^{\text{pred}}$) and the ground truth weights ($w_i^{\text{gt}}$).

$$\mathcal{L}_{\text{img}} = \frac{1}{N} \sum_{i=1}^{N} \|\text{rgb}_i - \text{gt\_rgb}_i\|_1$$

is the **image loss**, computed as the L1 loss between the predicted RGB image ($\text{rgb}_i$) and the novel view RGB image ($\text{gt\_rgb}_i$).

When using only the weight loss for supervision, $\lambda_{\text{weight}} = 1$ and $\lambda_{\text{img}} = 0$. When using only the image loss for supervision, $\lambda_{\text{weight}} = 0$ and $\lambda_{\text{img}} = 1$. When using both the weight and image loss for supervision, $\lambda_{\text{weight}} = 0.1$ and $\lambda_{\text{img}} = 1$.

Table 10: Human face reconstruction on CelebA-HQ.

| Method | HyPoGen | LDMI | Ours |
|---|---|---|---|
| SSIM ↑ | 0.4550 | 0.6947 | **0.8635** |
| LPIPS ↓ | 0.1463 | 0.0556 | **0.0296** |

### A.5 EXTENDED RESULTS

While the main paper focused on PSNR comparisons for clarity, here we present additional evaluation metrics to provide a more comprehensive assessment of reconstruction quality. Tables 10, 11, 12, 13 and 14 show SSIM and LPIPS results for CelebA-HQ, ImageNet, ERA5, ShapeNet Chairs and novel view synthesis on NeRF datasets, respectively.

**CelebA-HQ Results:** As shown in Table 10, our method achieves the highest SSIM score of 0.8635, significantly outperforming LDMI and HyPoGen. For LPIPS, our method demonstrates superior

perceptual quality with the lowest score of 0.0296, compared to LDMI (0.0556) and HyPoGen (0.1463), confirming better structural similarity and perceptual quality.

**ImageNet Results:** As shown in Table 11, our method achieves SSIM of 0.6064 and LPIPS of 0.1636, demonstrating competitive performance across diverse natural image domains.

Table 11: Natural image reconstruction on ImageNet.

| Method | VAMoH | HyPoGen | LDMI | Ours |
|---|---|---|---|---|
| SSIM ↑ | 0.5955 | 0.2473 | 0.5610 | **0.6064** |
| LPIPS ↓ | 0.4128 | 0.6141 | 0.1468 | **0.1636** |

**ERA5 Results:** As shown in Table 12, our method achieves near-perfect structural similarity (0.9991), marginally improving upon LDMI (0.9952) and HyPoGen (0.9947). For LPIPS, our method achieves the best perceptual quality (0.0001), outperforming LDMI (0.0003) and HyPoGen (0.0008).

Table 12: Climate data reconstruction on ERA5.

| Method | HyPoGen | LDMI | Ours |
|---|---|---|---|
| SSIM ↑ | 0.9947 | 0.9952 | **0.9991** |
| LPIPS ↓ | 0.0008 | 0.0003 | **0.0001** |

**ShapeNet Chairs Results:** As shown in Table 13, our method achieves the highest PSNR of 23.8 dB on rendered images, outperforming LDMI (22.4 dB) and HyPoGen (19.0 dB), demonstrating superior 3D shape reconstruction quality with more accurate geometry and texture representation.

**Pretrained NeRF Results:** As shown in Table 14, our method achieves the highest SSIM score of 0.948, outperforming LDMI (0.929) and HyPoGen (0.921). For LPIPS, our method obtains the best score of 0.0742, indicating superior perceptual quality compared to LDMI (0.1079) and HyPoGen (0.1182), highlighting effectiveness in generating high-quality novel views. Additionally, we include the results for Trans-INR (Chen & Wang, 2022) and GINR-IPC (Kim et al., 2022) on the Novel View Synthesis task using the Pretrained NeRF dataset. Here, for HyperDiffusion, HyPoGen, LDMI, and Ours, we use both the weight and image loss for supervision. As shown in Table 15, our method continues to outperform all other approaches, achieving the best performance across all metrics.

We also provide additional visual comparison in Figure 7-12.

### A.6 SCALABILITY ANALYSIS

To validate our framework's scalability advantage, we conduct experiments with varying amounts of training data (10%, 20%, 50%, and 100%) on pretrained NeRF, as illustrated in Figure 6. Our method consistently outperforms both baselines across all data scales, with several notable observations. First, all three methods exhibit steady performance improvements as training data increases, demonstrating the general benefit of larger datasets. However, the improvement trajectories differ significantly: our method shows a steeper and more consistent upward trend. This expanding gap indicates that our Transformer-based tokenization mechanism more effectively exploits additional training data compared to existing approaches.

Table 13: Results on ShapeNet Chairs.

| Method | HyPoGen | LDMI | Ours |
|---|---|---|---|
| PSNR on render Image | 19.0 | 22.4 | **23.8** |

### A.7 STABILITY ANALYSIS

To assess the robustness of our method with respect to random initialization, we perform a stability analysis across multiple random seeds on three diverse datasets: the Pretrained NeRF dataset,

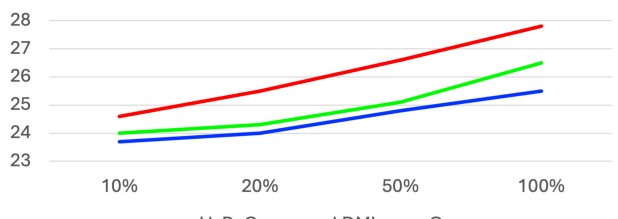

| scale | HyPoGen | LDMI | Ours |
|-------|---------|------|------|
| 10%   | 23.7    | 24.0 | **24.6** |
| 20%   | 24.0    | 24.3 | **25.5** |
| 50%   | 24.8    | 25.1 | **26.6** |
| 100%  | 25.5    | 26.5 | **27.8** |

Figure 6: Different amounts of training data on Pretrained NeRF.

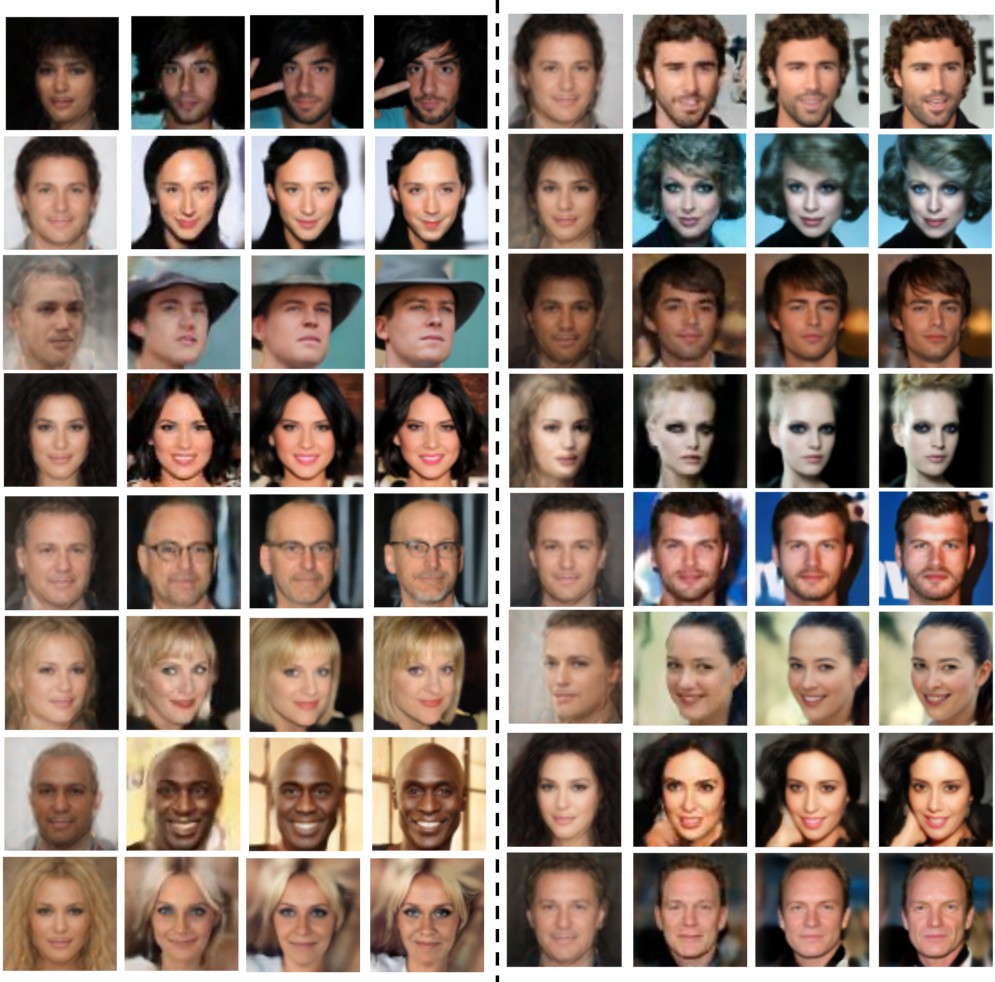

HyPoGen    LDMI    Ours    Input & GT | HyPoGen    LDMI    Ours    Input & GT

Figure 7: More visual samples on CelebA-HQ.

CelebA-HQ, and ERA5. For each dataset, we report the PSNR values obtained under different seeds, along with the corresponding mean and standard deviation.

As shown in Table 16 and 17, on the Pretrained NeRF dataset, the PSNR remains highly stable across four different seeds, fluctuating within a narrow range of 27.7–27.8, with an overall mean of 27.75 ± 0.06. Similarly, on CelebA-HQ, the variation across seeds is minimal, yielding an overall mean of 27.68 ± 0.13. On ERA5, the PSNR values range from 48.9 to 49.5, with a mean of 49.26 ±

Table 14: NeRF novel view synthesis on Pre-tained NeRF.

| Method | HyPoGen | LDMI | Ours |
|---|---|---|---|
| SSIM ↑ | 0.921 | 0.929 | **0.948** |
| LPIPS ↓ | 0.1182 | 0.1079 | **0.0742** |

Table 15: Novel View Synthesis on the Pretrained NeRF Dataset with More Baselines.

| Method | Trans-INR | GINR-IPC | HyperDiffusion | HyPoGen | LDMI | Ours |
|---|---|---|---|---|---|---|
| **PSNR (dB) ↑** | 21.9 | 26.1 | 18.1 | 25.5 | 26.5 | **27.8** |

0.23. These results demonstrate that our model exhibits strong consistency, even on image datasets with diverse appearance statistics.

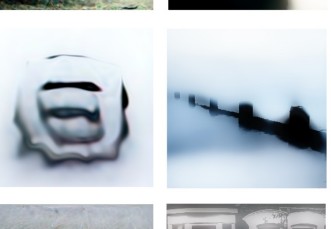
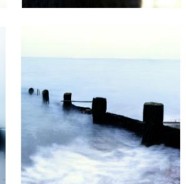
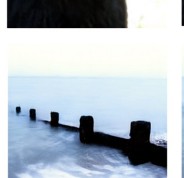
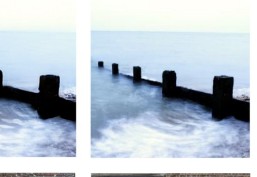
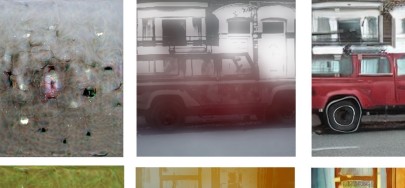
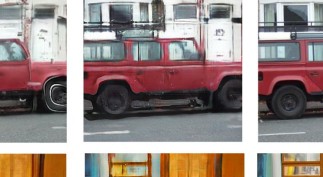
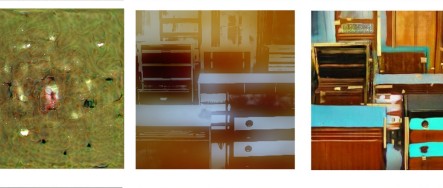
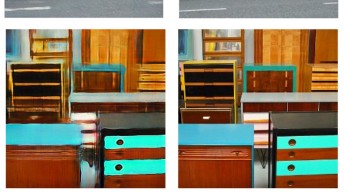
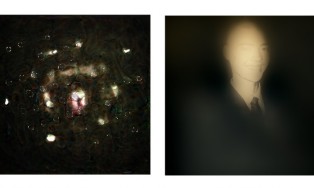
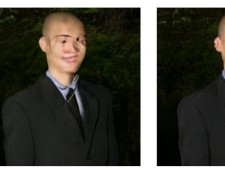
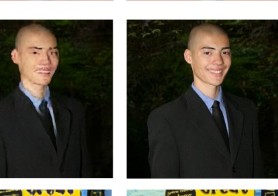
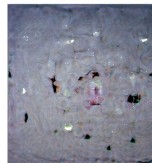
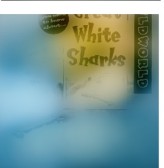
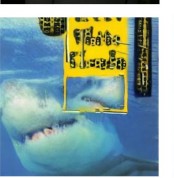
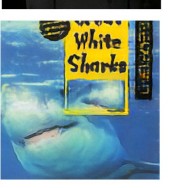
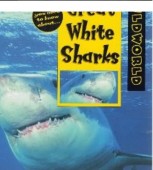

HyPoGen  VaMoH  LDMI  Ours  Input & GT

Figure 8: More visual samples on ImageNet.

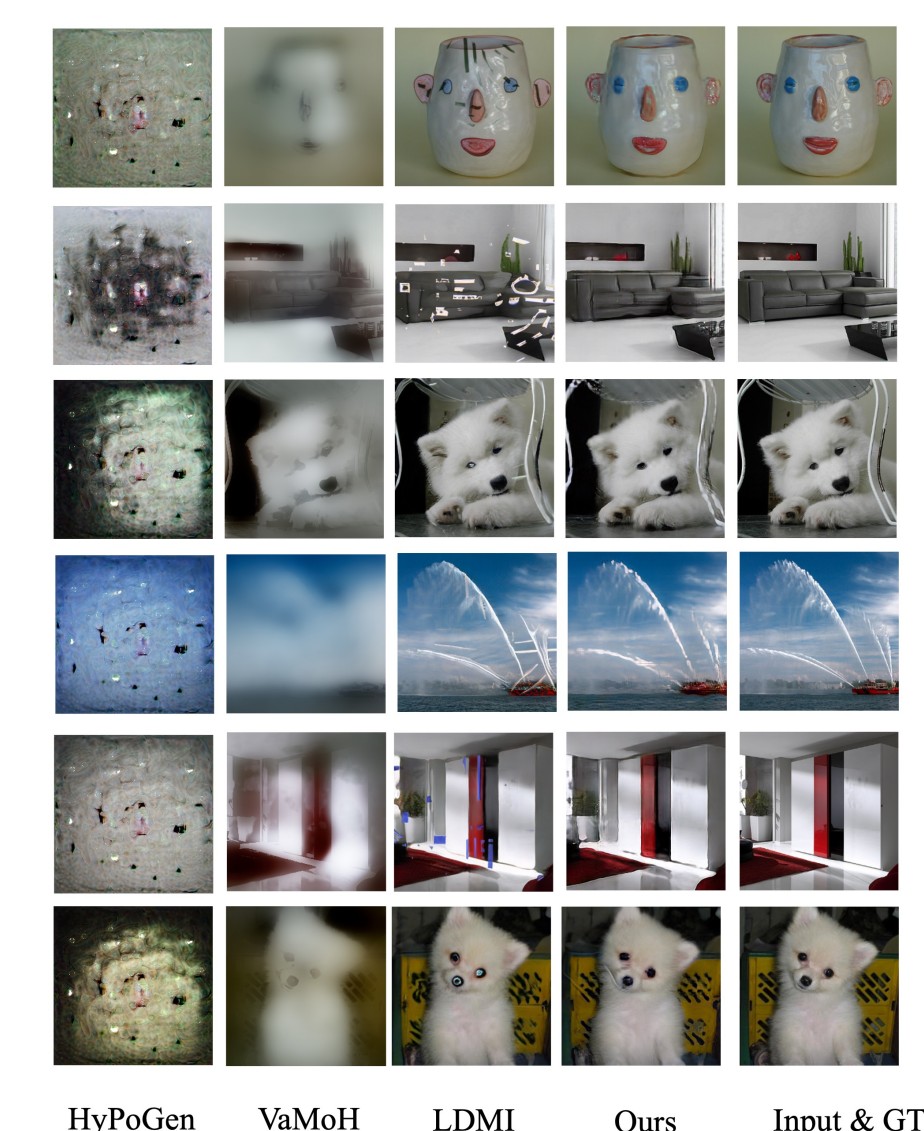

HyPoGen     VaMoH     LDMI     Ours     Input & GT

Figure 9: More visual samples on ImageNet.

Table 16: PSNR Across Different Random Seeds on the Pretrained NeRF and CelebA-HQ Datasets.

| Dataset | 444 | 555 | 777 | 888 | mean ± std |
|---|---|---|---|---|---|
| **Pretrained NeRF** | 27.7 | 27.8 | 27.7 | 27.8 | **27.75 ± 0.06** |
| **CelebA-HQ** | 27.8 | 27.7 | 27.7 | 27.5 | **27.68 ± 0.13** |

Table 17: PSNR Across Different Random Seeds on the ERA5 Dataset.

| Seed | 444 | 555 | 777 | 888 | 999 | mean ± std |
|---|---|---|---|---|---|---|
| **PSNR (dB) ↑** | 48.9 | 49.3 | 49.4 | 49.2 | 49.5 | **49.26 ± 0.23** |

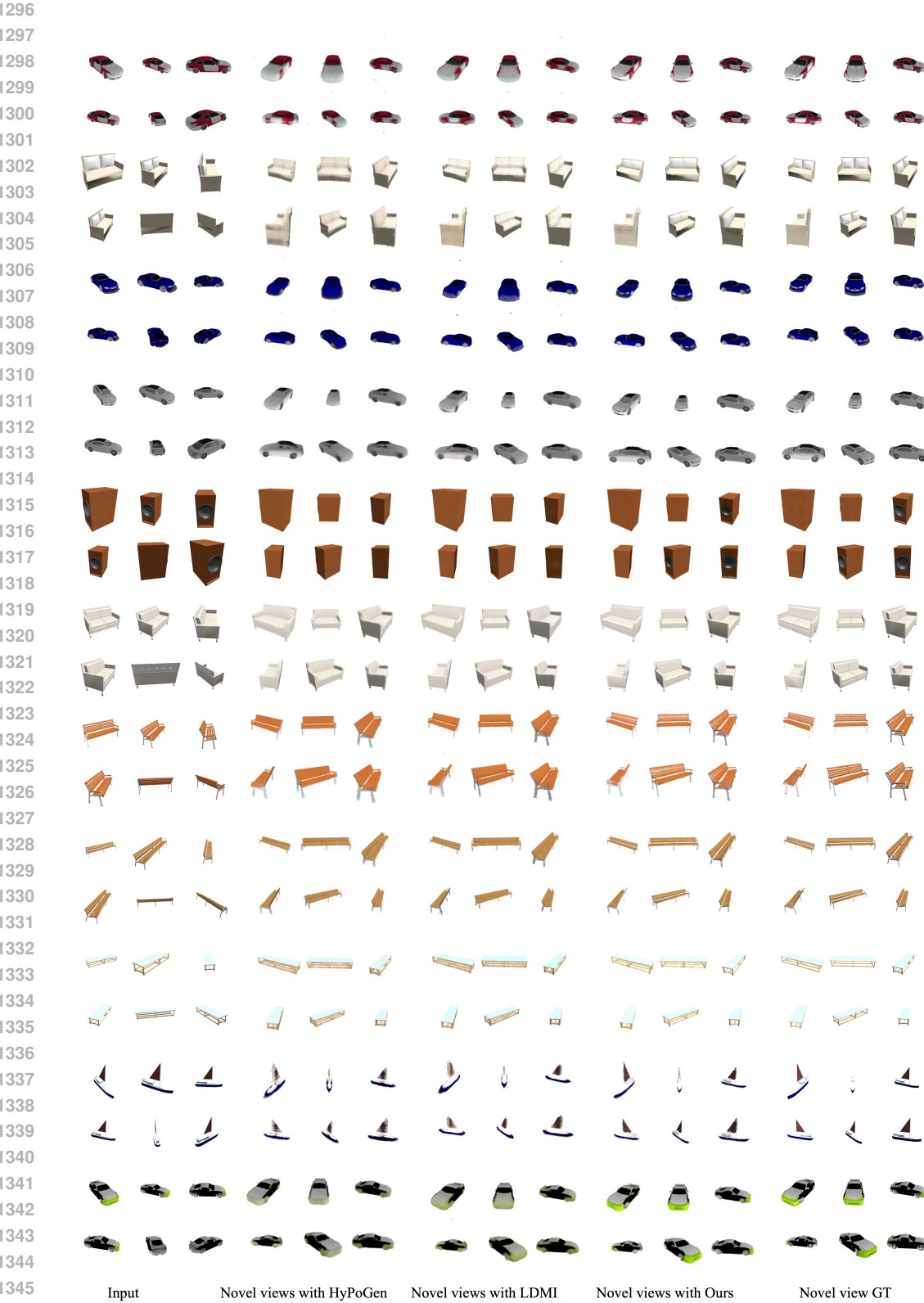

Input      Novel views with HyPoGen    Novel views with LDMI    Novel views with Ours    Novel view GT

Figure 10: More visual samples on NeRF.

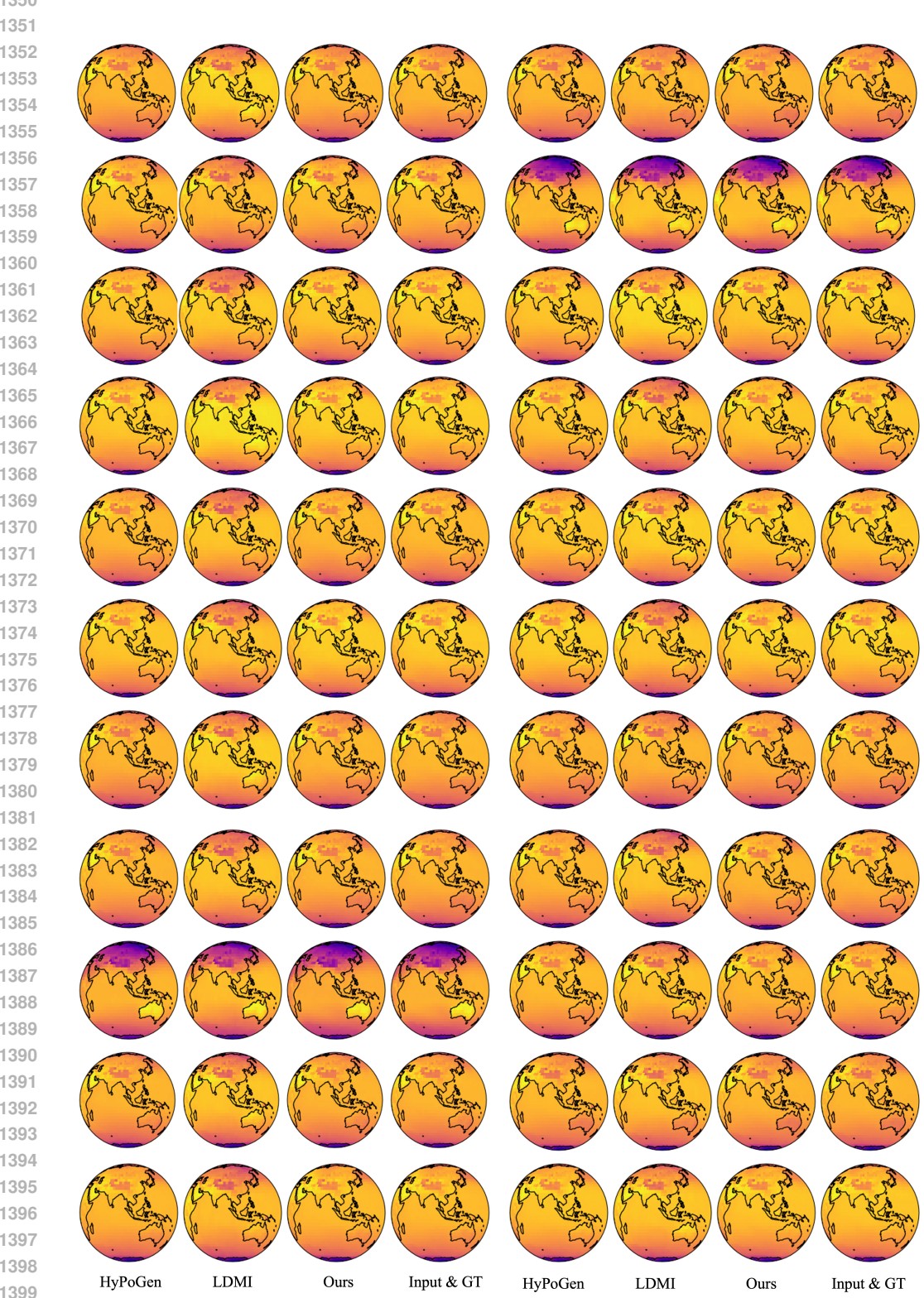

Figure 11: More visual samples on ERA5.

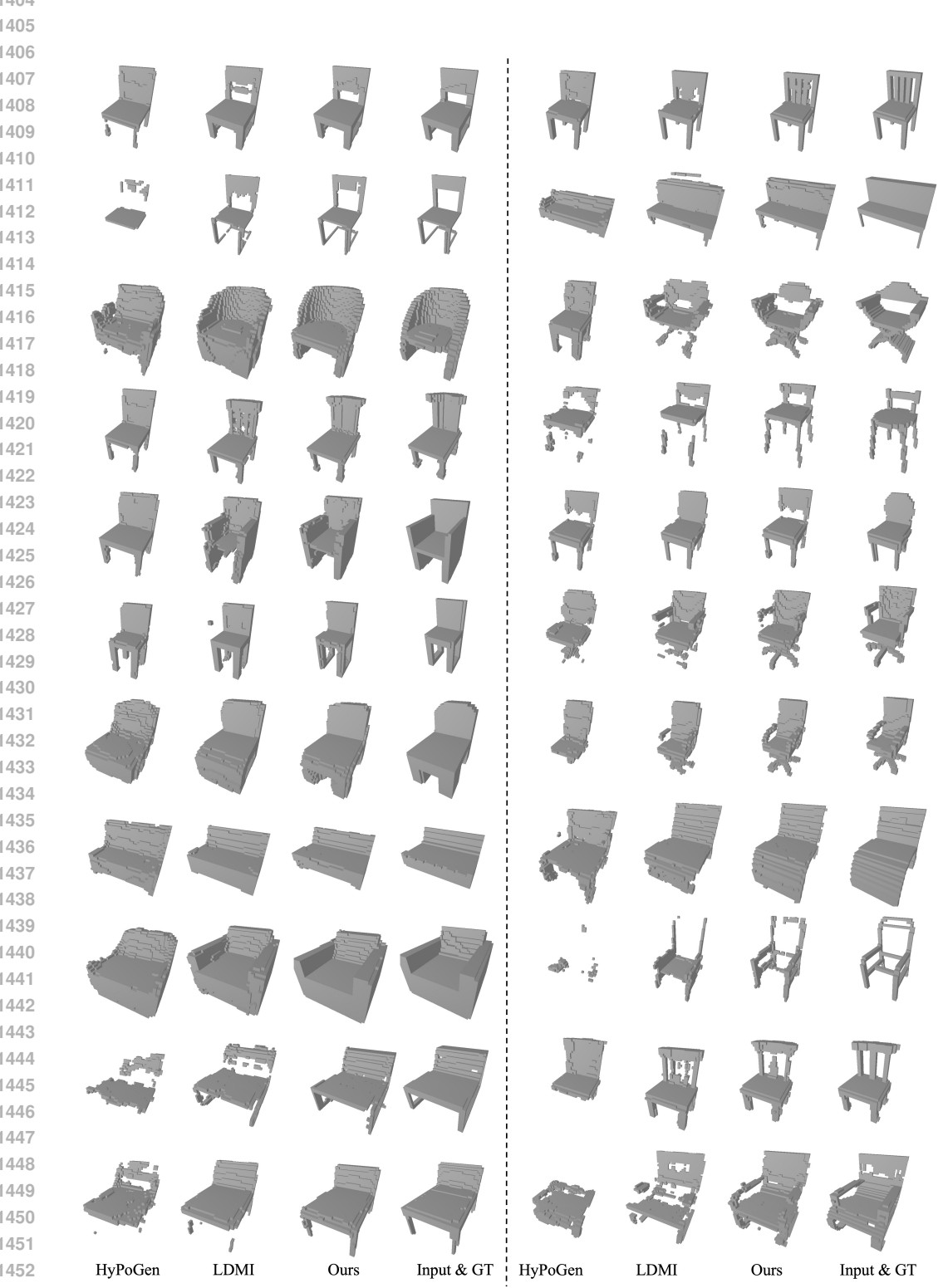

Figure 12: More visual samples on ShapeNet Chairs.

| HyPoGen | LDMI | Ours | Input & GT | HyPoGen | LDMI | Ours | Input & GT |

