# OpenReview forum: "Implicit Neural Representation Generation with Hypernetworks"
_ICLR.cc/2026/Conference — Submitted to ICLR 2026_

### Official Review · Reviewer_Q7Jz · 2025-10-21

**Soundness:** 3
**Presentation:** 2
**Contribution:** 3
**Rating:** 4
**Confidence:** 4

**Summary:**

This paper presents a Transformer-based hypernetwork framework for generating Implicit Neural Representations (INRs). The method addresses two key limitations of previous approaches: the lack of inter-layer dependency modeling and limited scalability. By viewing INR parameter generation as an optimization process, the authors employ cross-attention to model both the forward and backward gradient flows according to the chain rule, effectively capturing dependencies between layers.

The overall concept is interesting and theoretically sound, but the experimental evaluation is relatively weak, and the paper lacks sufficient implementation details to assess reproducibility fully.

**Strengths:**

1. Novel and well-motivated concept. The paper introduces an original idea that connects gradient-based optimization theory with Transformer attention mechanisms for INR generation. Modeling inter-layer dependencies through cross-attention and the chain rule is both elegant and theoretically justified.

2. Clear methodological design and firm theoretical grounding. The proposed architecture integrates attention-based gradient estimation, row-wise tokenization, and adaptive initialization coherently. The formulation is mathematically consistent and provides a principled alternative to previous MLP-based hypernetworks.

**Weaknesses:**

1. Limited experimental rigor. Although the reported results are promising, the experiments are relatively shallow. There is little statistical analysis, no mention of variance across runs, and no comparison of computational efficiency or training time with baselines.

2. Missing implementation details. Important architectural and training parameters are not fully specified, including the number of Transformer layers, embedding dimensions, token sequence lengths, and optimization hyperparameters. This makes reproducibility difficult.

3. Weak evaluation of scalability. The paper frequently claims scalability, but there is no empirical evidence such as runtime analysis, memory consumption, or scaling curves for large datasets or high-resolution signals.

4. Limited diversity of evaluation metrics. The experiments rely mainly on PSNR and simple accuracy measures. Additional metrics such as SSIM, Chamfer Distance, or LPIPS would provide a more complete evaluation, especially for 3D and visual quality tasks.

5. Overfitting to specific architectures. The adaptive initialization and SIREN-based design seem tuned to a particular INR architecture. It is unclear whether the proposed method generalizes to other types of implicit networks, such as ReLU or Fourier-feature-based models.

6. Presentation and clarity issues in the experimental section. The visual presentation of the results is weak. The figures are poorly designed and do not effectively support the claims made in the text. Figure 1 adds little value, as it does not clearly illustrate the proposed method or its components. Figure 4 shows very low-quality reconstructions, suggesting that the model may be undertrained or not properly converged. Figure 3 is poorly described and difficult to interpret, with unclear labeling and low visual readability. Overall, the experimental figures fail to convincingly demonstrate the strengths of the proposed approach.

**Questions:**

1. Can the authors provide statistical measures (e.g., mean ± std over multiple seeds) to assess the stability of training?
2. How does the proposed method compare to baselines in terms of computational efficiency, convergence speed, or training time?
3. Is the code (or sufficient pseudocode) available to enable reproducibility of the reported results?
4. What empirical evidence supports the claimed scalability of the approach?
5. Why are only PSNR or simple accuracy measures used?
6. Could additional perceptual or geometric metrics (e.g., SSIM, LPIPS, Chamfer Distance) provide a more comprehensive evaluation, particularly for 3D or visual reconstruction tasks?
7. Does the method generalize beyond SIREN-based INRs?

---

> ### Author Response · Authors · 2025-11-23
>
> # Response to Reviewer Q7Jz
>
> Dear Reviewer Q7Jz,
>
> Thank you for your valuable feedback. We are delighted to hear that you found our method a "novel and well-motivated concept" that is "elegant and theoretically justified," with a "clear methodological design and firm theoretical grounding." We address your concerns and comments below:
>
> ---
>
> ## A to W1 & Q1 & Q2: Statistical Analysis, Stability, and Computational Resources
>
> > **Reviewer's Comment**: The experiments lack depth, with insufficient statistical analysis (e.g., variance across runs), no stability measures over multiple seeds, and absent comparisons of computational efficiency, convergence speed, or training time against baselines.
>
> ### Stability Across Random Seeds
>
> Due to computational and time constraints, we were not able to run a large number of seeds. However, on the **Pretrained NeRF dataset**, we trained our model with two different seeds to assess stability. The results are highly consistent, with minimal variance:
>
> ### PSNR Across Different Random Seeds on the Pretrained NeRF Dataset
>
> | **Seed** | **PSNR** |
> | -------- | -------- |
> | 555      | 27.8     |
> | 777      | 27.7     |
>
> ### Runtime Analysis
>
> In response to your request, we have included comparisons of **training time**, **memory usage**, and **inference speed** between our method and the baseline methods (HyPoGen and LDMI). These comparisons are provided in the tables below, which we hope will clarify the computational costs and advantages of our method.
>
> ### Comparison of Train Time, Memory Consumption, and Model Size Across Datasets
>
> | Dataset             | Throughput (sample/s) |      |      | GPU Memory (GB) / Max Batch Size |             |             | Params (M) |       |       |
> | ------------------- | --------------------- | ---- | ---- | -------------------------------- | ----------- | ----------- | ---------- | ----- | ----- |
> |                     | HyPoGen               | LDMI | Ours | HyPoGen                          | LDMI        | Ours        | HyPoGen    | LDMI  | Ours  |
> | **CelebA-HQ**       | 630                   | 600  | 213  | 73.32 / 900                      | 77.71 / 420 | 78.21 / 256 | 128.8      | 40.5  | 41.2  |
> | **ImageNet**        | 142                   | 52   | 70   | 75.97 / 60                       | 74.35 / 40  | 78.12 / 42  | 578.7      | 544.9 | 124.5 |
> | **ERA5**            | 3111                  | 616  | 170  | 77.68 / 1400                     | 72.97 / 128 | 78.24 / 252 | 93.1       | 48.9  | 21.3  |
> | **Shapes3D**        | 2660                  | 1500 | 806  | 74.91 / 700                      | 74.79 / 420 | 73.50 / 576 | 8.3        | 30.9  | 4.1   |
> | **Pretrained NeRF** | 15                    | 16   | 30   | 76.03 / 56                       | 79.02 / 16  | 75.58 / 40  | 1049.9     | 515.5 | 482.5 |
>
> ### Comparison of Test Time and Memory Consumption with Batch Size Set to 1
>
> | Dataset             | Test-time GPU Memory (GB) |       |       | Inference Time (ms) |      |      |
> | ------------------- | ------------------------- | ----- | ----- | ------------------- | ---- | ---- |
> |                     | HyPoGen                   | LDMI  | Ours  | HyPoGen             | LDMI | Ours |
> | **CelebA-HQ**       | 1.037                     | 0.854 | 0.389 | 5                   | 18   | 166  |
> | **ImageNet**        | 4.757                     | 10.32 | 1.382 | 8                   | 27   | 89   |
> | **ERA5**            | 0.769                     | 0.285 | 0.238 | 9                   | 13   | 81   |
> | **Shapes3D**        | 0.172                     | 0.138 | 0.142 | 3                   | 9    | 55   |
> | **Pretrained NeRF** | 9.286                     | 7.43  | 6.08  | 169                 | 167  | 205  |
>
> ---
>
> ##

---

> > ### Author Response · Authors · 2025-11-23
> >
> > ## A to W2 & Q3: Hyperparameter Specifications and Reproducibility
> >
> > > **Reviewer's Comment**: The paper lacks specification of critical architectural (e.g., Transformer layers, embedding dimensions, token lengths) and training hyperparameters, impeding reproducibility, and provides no information on code or pseudocode availability.
> >
> > As requested, we have provided full details of the **token sequence length** for each Linear layer:
> >
> > #### TokenSeqLen = out_dim + (out_dim / in_dim).
> >
> > The first out_dim tokens correspond to the **row-wise** grouping of the weight matrix **W** (each token is **in_dim-dimensional**), while the remaining **out_dim/in_dim** tokens correspond to the bias **b**, grouped into **in_dim-dimensional chunks**.
> >
> > Due to space constraints, we provide full details in **Appendix A.2.1** and **Appendix A.3.2**, including the number of Transformer layers, embedding dimensions, token sequence lengths, and optimization hyperparameters. We have revised these details in the updated version and will release our code to ensure reproducibility and support future research.
> >
> > ### Model Hyperparameters Across Different Datasets
> >
> > | Parameter                           | CelebA-HQ 64×64   | ImageNet          | ERA5              | Chairs    | Pre-trained NeRF |
> > | ----------------------------------- | ----------------- | ----------------- | ----------------- | --------- | ---------------- |
> > | **Learning rate**                   | 5e-5              | 5e-5              | 5e-5              | 5e-5      | 5e-5             |
> > | **Optimizer**                       | Adam              | Adam              | Adam              | Adam      | Adam             |
> > | **Mini-batch size**                 | 64                | 128               | 128               | 32        | 96               |
> > | **Task Tokenizer dim**              | 128               | 256               | 128               | 64        | 768              |
> > | **Each Block Hidden dim**           | 128               | 768               | 128               | 64        | 768              |
> > | **Transformer layers**              | 1                 | 1                 | 1                 | 1         | 1                |
> > | **Token len**                       | 384,257,257,257,4 | 384,257,257,257,4 | 342,257,257,257,2 | 171,129,2 | 64,64,64,16      |
> > | **INR type**                        | SIREN             | SIREN             | SIREN             | SIREN     | NeRF             |
> > | **INR layers**                      | 5                 | 5                 | 5                 | 3         | 3                |
> > | **INR hidden dim**                  | 256               | 256               | 256               | 128       | 64               |
> > | **INR $\omega$ / encode frequency** | 30                | 30                | 30                | 30        | 24               |
> >
> > ---
> >
> > ## A to W3 & Q4: Scalability and Performance on Large Datasets
> >
> > > **Reviewer's Comment**: The paper's claims of scalability are unsubstantiated by scaling performance on large datasets or high-resolution signals.
> >
> > In response to this concern, we conducted experiments with varying amounts of training data. The results are reported below. Our method consistently outperforms the baselines, especially as the dataset size increases. We plan to update these results with different network parameter scales as soon as resources allow.
> >
> > ### Different Amounts of Training Data on Pretrained NeRF
> >
> > | Scale     | HyPoGen | LDMI | Ours     |
> > | --------- | ------- | ---- | -------- |
> > | **10 %**  | 23.7    | 24.0 | **24.6** |
> > | **20 %**  | 24.0    | 24.3 | **25.5** |
> > | **50 %**  | 24.8    | 25.1 | **26.6** |
> > | **100 %** | 25.5    | 26.5 | **27.8** |
> >
> > ---
> >
> > ##

---

> > > ### Author Response · Authors · 2025-11-23
> > >
> > > ## A to W4 & Q5 & Q6: Evaluation Metrics and Additional Comparisons
> > >
> > > > **Reviewer's Comment**: The evaluation metrics are overly reliant on PSNR and basic accuracy, lacking diversity; incorporating additional perceptual and geometric metrics (e.g., SSIM, LPIPS) is recommended for a more robust assessment of 3D and visual reconstruction quality.
> > >
> > > Due to page limitations, we focused on one representative evaluation metric in the main paper. However, we have included additional evaluation metrics such as **SSIM**, **LPIPS** in the revised manuscript to provide a more comprehensive assessment of our method.
> > >
> > > ### Human Face Reconstruction on CelebA-HQ
> > >
> > > | Method      | HyPoGen | LDMI   | Ours       |
> > > | ----------- | ------- | ------ | ---------- |
> > > | **SSIM ↑**  | 0.4550  | 0.6947 | **0.8635** |
> > > | **LPIPS ↓** | 0.1463  | 0.0556 | **0.0296** |
> > >
> > > ### Natural Image Reconstruction on ImageNet
> > >
> > > | Method      | VAMoH  | HyPoGen | LDMI   | Ours       |
> > > | ----------- | ------ | ------- | ------ | ---------- |
> > > | **SSIM ↑**  | 0.5955 | 0.2473  | 0.5610 | **0.6064** |
> > > | **LPIPS ↓** | 0.4128 | 0.6141  | 0.1468 | **0.1636** |
> > >
> > > ### Climate Data Reconstruction on ERA5
> > >
> > > | Method      | HyPoGen | LDMI   | Ours       |
> > > | ----------- | ------- | ------ | ---------- |
> > > | **SSIM ↑**  | 0.9947  | 0.9952 | **0.9991** |
> > > | **LPIPS ↓** | 0.0008  | 0.0003 | **0.0001** |
> > >
> > > ### Results on ShapeNet Chairs
> > >
> > > | Method                   | HyPoGen | LDMI | Ours     |
> > > | ------------------------ | ------- | ---- | -------- |
> > > | **PSNR on Render Image** | 19.0    | 22.4 | **23.8** |
> > >
> > > ### NeRF Novel View Synthesis on Pretrained NeRF
> > >
> > > | Method      | HyPoGen | LDMI   | Ours                                          |
> > > | ----------- | ------- | ------ | --------------------------------------------- |
> > > | **SSIM ↑**  | 0.921   | 0.929  | <span style="font-weight:bold;">0.948</span>  |
> > > | **LPIPS ↓** | 0.1182  | 0.1079 | <span style="font-weight:bold;">0.0742</span> |
> > >
> > > ---
> > >
> > > ## A to W5 & Q7: Generalization Beyond SIREN
> > >
> > > > **Reviewer's Comment**: The method's adaptive initialization and SIREN-specific design raise concerns about overfitting to particular INR architectures, with unclear evidence of generalization to alternatives like ReLU or Fourier-feature-based models.
> > >
> > > While the adaptive initialization was empirically designed for **SIREN-based networks**, our method generalizes well to other **implicit networks**. In our NeRF experiments, we used a **ReLU-based INR** instead of SIREN to verify its general effectiveness. The results show that our method works well across different types of INRs.
> > >
> > > ---
> > >
> > > ## A to W6: Presentation and Clarity of Figures
> > >
> > > > **Reviewer's Comment**: The experimental section exhibits significant presentation and clarity deficiencies, with poorly designed figures that inadequately support textual claims, including a non-informative Figure 1, low-quality and potentially undertrained reconstructions in Figure 4, and a confusingly labeled, low-readability Figure 3.
> > >
> > > We sincerely apologize for the confusion caused by the figures. We have made the following improvements:
> > >
> > > - **Figure 1** now clearly showcases diverse samples from various tasks, demonstrating the versatility of our proposed method.
> > > - **Figure 3** has been redesigned to include clearer labels and improved readability, with additional captions for each part of the figure.
> > > - **Figure 4** has been updated to show the **sample supervision** results, replacing the original weight supervision-based results. We have also improved the visualizations and added more examples in the appendix.
> > >
> > > ---
> > >
> > > We hope these revisions address your concerns. Thank you once again for your valuable feedback, which has greatly improved the quality of our manuscript. Please feel free to let us know if you have further questions.

---

> > > > ### Comment · Reviewer_Q7Jz · 2025-11-25
> > > >
> > > > Thank you for the detailed response and the updated paper. While the efficiency tables and hyperparameter details are helpful, I have three remaining concerns:
> > > >
> > > > 1. Stability Analysis
> > > >
> > > > Results from only two seeds are insufficient for statistical analysis. To truly prove robustness, please provide mean ± std over at least 3-5 seeds, even if performed on a smaller dataset or reduced model.
> > > >
> > > > 2. Scalability vs. Speed
> > > >
> > > > The new tables show that your method is parameter-efficient but suffers from a significant computational bottleneck. Please clarify how you define scalability, given this trade-off; currently, the method scales well with respect to parameters but poorly with respect to runtime.
> > > >
> > > > 3. Code Availability
> > > >
> > > > Given that ICLR is an open venue and significant time has passed since the initial submission, I do not understand why the code has not been made available yet. Reliance on a promise to release code "upon acceptance" is insufficient for verifying reproducibility during the review process, especially for a complex architecture with custom tokenization.

---

> > > > > ### Author Response · Authors · 2025-11-26
> > > > >
> > > > > # Response to Reviewer Q7Jz
> > > > >
> > > > > Dear Reviewer Q7Jz,
> > > > >
> > > > > Thank you for the insightful comment.
> > > > >
> > > > > ---
> > > > >
> > > > > ## A to Q1: Stability Analysis
> > > > >
> > > > > > **Reviewer's Comment**: Results from only two seeds are insufficient for statistical analysis. To truly prove robustness, please provide mean ± std over at least 3-5 seeds, even if performed on a smaller dataset or reduced model.
> > > > >
> > > > > We have further strengthened the stability analysis by expanding the number of random seeds across multiple datasets.  Specifically, on the **Pretrained NeRF Dataset**, we added **two additional seeds**, yielding a performance of 27.75 ± 0.06 across four seeds.  In addition, we updated the results on **CelebA-HQ** using **four** seeds (27.68 ± 0.13) and on **ERA5** using **five** seeds (49.26 ± 0.23).  The consistently low variance across all datasets confirms that our implementation is robust and not sensitive to random initialization.
> > > > >
> > > > > ### PSNR Across Different Random Seeds on the Pretrained NeRF Dataset
> > > > >
> > > > > | **Seed**   | **PSNR**         |
> > > > > | ---------- | ---------------- |
> > > > > | 444        | 27.7             |
> > > > > | 555        | 27.8             |
> > > > > | 777        | 27.7             |
> > > > > | 888        | 27.8             |
> > > > > | mean ± std | **27.75 ± 0.06** |
> > > > >
> > > > > ### PSNR Across Different Random Seeds on CelebA-HQ
> > > > >
> > > > > | **Seed**   | **PSNR**         |
> > > > > | ---------- | ---------------- |
> > > > > | 444        | 27.8             |
> > > > > | 555        | 27.7             |
> > > > > | 777        | 27.7             |
> > > > > | 888        | 27.5             |
> > > > > | mean ± std | **27.68 ± 0.13** |
> > > > >
> > > > > ### PSNR Across Different Random Seeds on ERA5
> > > > >
> > > > > | **Seed**   | **PSNR**         |
> > > > > | ---------- | ---------------- |
> > > > > | 444        | 48.9             |
> > > > > | 555        | 49.3             |
> > > > > | 777        | 49.4             |
> > > > > | 888        | 49.2             |
> > > > > | 999        | 49.5             |
> > > > > | mean ± std | **49.26 ± 0.23** |
> > > > >
> > > > >
> > > > >
> > > > > ## A to Q2: Scalability vs. Speed
> > > > >
> > > > > > **Reviewer's Comment**: The new tables show that your method is parameter-efficient but suffers from a significant computational bottleneck. Please clarify how you define scalability, given this trade-off; currently, the method scales well with respect to parameters but poorly with respect to runtime.
> > > > >
> > > > > Thank you very much for the thoughtful observation. We appreciate the opportunity to clarify our definition of scalability. In our paper, **scalability specifically refers to data scalability**. That is, under **the same parameter budget**, our method exhibits **larger performance gains as the training data scale increases**, compared to existing approaches. This trend is clearly demonstrated in **Figure 6** in the current revision, where our method continues to improve as more training samples are available, while baselines begin to saturate much earlier.
> > > > >
> > > > > As reported in Table **Comparison of Train Time, Memory Consumption, and Model Size Across Datasets**, the training-time throughput and memory consumption of our method are broadly comparable to the baselines, indicating that our approach does not introduce additional overhead during training while maintaining the same parameter budget.
> > > > >
> > > > > Regarding Table **Comparison of Test Time and Memory Consumption with Batch Size Set to 1**, we acknowledge that our method shows a somewhat higher test-time latency.  However, in typical NeRF usage scenarios, a model is generated once, and the subsequent computational cost is primarily dominated by multi-view rendering, which remains the main runtime factor independent of our method.  Therefore, the inference-time difference has limited practical impact, as the task is generally not time-critical.
> > > > >
> > > > > ## A to Q3: Code Availability
> > > > >
> > > > > > **Reviewer's Comment**: Given that ICLR is an open venue and significant time has passed since the initial submission, I do not understand why the code has not been made available yet. Reliance on a promise to release code "upon acceptance" is insufficient for verifying reproducibility during the review process, especially for a complex architecture with custom tokenization.
> > > > >
> > > > > Thank you for pointing this out, and we sincerely apologize for the delayed release.  We have now included the codebase in the supplementary materials to ensure full reproducibility during the review process.

---

### Official Review · Reviewer_3wHy · 2025-10-22

**Soundness:** 2
**Presentation:** 2
**Contribution:** 2
**Rating:** 4
**Confidence:** 4

**Summary:**

This paper proposes to improve implicit neural representation (INR) generation from hypernetworks by improving on HyPoGen. HyPoGen generates the weights of an INR by mimicking backward passes (i.e. INR optimization) with neural network modules. This paper improves on HyPoGen in three ways: 1) using an attention-based module to mimic gradient updates, 2) a way to tokenize INR parameters for input, and 3) an initialization strategy to stabilize training when the INR whose weights are being learned is SIREN. This paper evaluates on generation INRs for 2D images (CelebA-HQ, ImageNet), climate data (ERA5), and 3D data (ShapeNet chairs, NeRF dataset).

**Strengths:**

This paper proposes a new hypernetwork architecture, based on HyPoGen, inspired by the idea that representing the “backward pass” as a neural network module can be done with attention. The evaluation is done on several different tasks and datasets.

**Weaknesses:**

**(W1) Novelty**: This method makes incremental improvements over HyPoGen, on which it is based.

**(W2) Clarity**: From the writing, it is hard to understand how a forward pass of the model works, and it’s also hard to understand how the learned weights are ultimately used. It is also not very clear what loss functions (e.g. sample-supervised or weight-supervised) are used to train the models for each of the datasets.

**(W3) Empirical evaluation**: This paper is missing baselines on some tasks. For novel view synthesis, there are many works using hypernetworks to accomplish this task [1-5], using a variety of different approaches.

Additionally, it is difficult to compare approaches because there is no comparison of the number of model parameters between models. With some of the models using the proposed method being quite large (up to 480M parameters, for the novel view synthesis task), it raises the possibility that the proposed method is performing better due to being much larger than the other models. Similarly, there is no indication that each of the models is using the same size INR, which could also improve results without demonstrating the superiority of the proposed method.

I also don’t understand why accuracy is used to evaluate performance on 3D INR generation with ShapeNet chairs. Why not use PSNR, SSIM, LPIPS, FID? This metric seems less than ideal because it may reward reproducing all the background pixels correctly while not correctly rewarding reproducing the shape of the chair.

**(W4) Potential limitations**: Related to the above limitation, it’s not clear if the proposed model needs to be supervised on the weights for good performance (c.f. Table 3). This is a major limitation due to the need to collect weight datasets for training.

[1] Chen, Yinbo, and Xiaolong Wang. "Transformers as meta-learners for implicit neural representations." European Conference on Computer Vision. Cham: Springer Nature Switzerland, 2022.

[2] Kim, Chiheon, et al. "Generalizable implicit neural representations via instance pattern composers." Proceedings of the IEEE/CVF Conference on Computer Vision and Pattern Recognition. 2023.

[3] Gu, Jeffrey, Kuan-Chieh Wang, and Serena Yeung. "Generalizable neural fields as partially observed neural processes." Proceedings of the IEEE/CVF International Conference on Computer Vision. 2023.

[4] Lee, Doyup, et al. "Locality-aware generalizable implicit neural representation." Advances in Neural Information Processing Systems 36 (2023): 48363-48381.

[5] Gu, Jeffrey, and Serena Yeung-Levy. "Foundation models secretly understand neural network weights: Enhancing hypernetwork architectures with foundation models." arXiv preprint arXiv:2503.00838 (2025).

**Questions:**

**(Q1)**: What are the computational resources used by the proposed method and baseline methods?

**(Q2)**: Do all methods use the same size INRs and the same conditions (where relevant)?

**(Q3)**: How does a full forward pass of the network proceed?

**(Q4)**: Why is accuracy used instead of PSNR (or similar metrics) to evaluate performance on ShapeNet Chairs (Table 2)?

**(Q5)**: How does the performance of image-only, weight-only, and image + weight supervision compare?

**(Q6)**: For Table 3 (NeRF dataset), what is the performance of hypernetworks compared to the pre-trained NeRFs?

---

> ### Author Response · Authors · 2025-11-23
>
> # Response to Reviewer 3wHy
>
> Dear Reviewer 3wHy,
>
> Thank you for your thoughtful feedback and for recognizing our **new hypernetwork architecture** and evaluations across several tasks and datasets. We address your concerns below:
>
> ---
>
> ## A to W1: Incremental Improvements Over HyPoGen
>
> > **Reviewer's Comment**: This method makes incremental improvements over HyPoGen, on which it is based.
>
> Rather than pursuing groundbreaking concepts, our work aims to develop a **practical solution** that combines **inter-layer dependencies** with **scalable architectures**. While we share a similar approach to inter-layer dependencies with HyPoGen, our methodology differs significantly. We are the first to successfully integrate inter-layer dependencies into the **Transformer architecture**, through three non-trivial technical contributions:
>
> 1. **Attention-based gradient estimation**.
> 2. A carefully designed **tokenization mechanism**.
> 3. An effective **initialization strategy**.
>
> These contributions allow our model to scale effectively to larger datasets, a limitation observed in earlier hypernetwork-based INR generation methods like HyPoGen. This integration of **inter-layer dependencies** in the **Transformer architecture** introduces a substantial shift in how we model neural network weights, providing a unique advantage over prior work.
>
> ---
>
> ## A to W2 & Q3: Clarifications on Forward Pass, Weights, and Loss Functions
>
> > **Reviewer's Comment**: The paper lacks clarity on the model's forward pass mechanism, the application of learned weights, and the specific loss functions (e.g., sample- vs. weight-supervised) employed for training on each dataset.
>
> We appreciate your questions and have revised the manuscript to provide a clearer explanation of the **forward pass** and **loss functions**. Here’s a detailed clarification:
>
> - **Forward Pass**: Given a **task embedding**, the forward pass begins with the **token initialization network** (detailed in A.2.1). This network generates the initial tokens, which represent the weights **θ** of the target network. These tokens are then passed to the **hypernetwork**, which updates the weights iteratively by outputting **δθ**. These updated weights are subsequently used in the target network (**INRs**) for the forward pass as usual.
>
> - **Supervision Type**: For the **NeRF dataset**, we conducted experiments with both **weight-supervised** and **weight+sample-supervised** training. For all other datasets, we use **sample-supervised training**, as pre-trained weights are generally not available for most real-world tasks.
>
> We have now included the full experimental details, including the specific loss functions applied to each dataset, in the manuscript.
>
> ### Novel View Synthesis on NeRF Dataset
>
> | Supervision        | HyperDiffusion | HyPoGen | LDMI | Ours     |
> | ------------------ | -------------- | ------- | ---- | -------- |
> | **Weight only**    | 20.0           | 19.9    | 19.3 | **22.0** |
> | **Image only**     | *Not required* | 24.9    | 26.2 | **28.8** |
> | **Weight & Image** | **18.1**       | 25.5    | 26.5 | **27.8** |
>
> ---
>
> ##

---

> > ### Author Response · Authors · 2025-11-23
> >
> > ## A to W3: Model Parameter Comparisons, INR Size, and Evaluation Metrics
> >
> > > **Reviewer's Comment**: The empirical evaluation lacks key baselines for novel view synthesis, omits parameter and INR size comparisons that could explain performance differences due to model scale, and uses an inadequate accuracy metric for ShapeNet chairs that may favor background fidelity over shape reconstruction.
> >
> > Thank you for your feedback. We have revised the manuscript to provide additional comparisons of **model parameters** and **INR size** for all methods, ensuring fair comparisons across datasets and tasks.
> >
> > ### Model Parameter Scale (Q2)
> >
> > We report the model parameters and hyperparameters for each dataset in the following table. This ensures that all methods are evaluated using the same size **INRs** under the same conditions.
> >
> > ### Model Hyperparameters Across Different Datasets
> >
> > | Parameter                           | CelebA-HQ 64×64   | ImageNet          | ERA5              | Chairs    | Pre-trained NeRF |
> > | ----------------------------------- | ----------------- | ----------------- | ----------------- | --------- | ---------------- |
> > | **Learning rate**                   | 5e-5              | 5e-5              | 5e-5              | 5e-5      | 5e-5             |
> > | **Optimizer**                       | Adam              | Adam              | Adam              | Adam      | Adam             |
> > | **Mini-batch size**                 | 64                | 128               | 128               | 32        | 96               |
> > | **Task Tokenizer dim**              | 128               | 256               | 128               | 64        | 768              |
> > | **Each Block Hidden dim**           | 128               | 768               | 128               | 64        | 768              |
> > | **Transformer layers**              | 1                 | 1                 | 1                 | 1         | 1                |
> > | **Token len**                       | 384,257,257,257,4 | 384,257,257,257,4 | 342,257,257,257,2 | 171,129,2 | 64,64,64,16      |
> > | **INR type**                        | SIREN             | SIREN             | SIREN             | SIREN     | NeRF             |
> > | **INR layers**                      | 5                 | 5                 | 5                 | 3         | 3                |
> > | **INR hidden dim**                  | 256               | 256               | 256               | 128       | 64               |
> > | **INR $\omega$ / encode frequency** | 30                | 30                | 30                | 30        | 24               |
> >
> > ### Comparison of Train Time, Memory Consumption, and Model Size Across Datasets
> >
> > | Dataset             | Throughput (sample/s) |      |      | GPU Memory (GB) / Max Batch Size |             |             | Params (M) |       |       |
> > | ------------------- | --------------------- | ---- | ---- | -------------------------------- | ----------- | ----------- | ---------- | ----- | ----- |
> > |                     | HyPoGen               | LDMI | Ours | HyPoGen                          | LDMI        | Ours        | HyPoGen    | LDMI  | Ours  |
> > | **CelebA-HQ**       | 630                   | 600  | 213  | 73.32 / 900                      | 77.71 / 420 | 78.21 / 256 | 128.8      | 40.5  | 41.2  |
> > | **ImageNet**        | 142                   | 52   | 70   | 75.97 / 60                       | 74.35 / 40  | 78.12 / 42  | 578.7      | 544.9 | 124.5 |
> > | **ERA5**            | 3111                  | 616  | 170  | 77.68 / 1400                     | 72.97 / 128 | 78.24 / 252 | 93.1       | 48.9  | 21.3  |
> > | **Shapes3D**        | 2660                  | 1500 | 806  | 74.91 / 700                      | 74.79 / 420 | 73.50 / 576 | 8.3        | 30.9  | 4.1   |
> > | **Pretrained NeRF** | 15                    | 16   | 30   | 76.03 / 56                       | 79.02 / 16  | 75.58 / 40  | 1049.9     | 515.5 | 482.5 |
> >
> > ---
> >
> > ### Evaluation Metrics on ShapeNet (Q4)
> >
> > > **Reviewer's Comment**: The reviewer raises concerns about the evaluation metric for ShapeNet chairs, suggesting it may favor background fidelity over shape reconstruction.
> >
> > We use **occupancy accuracy** as the evaluation metric for 3D geometry reconstruction, which we believe more accurately captures the structural fidelity of the 3D shapes. However, we have also included **PSNR of rendered images** to demonstrate that our method consistently outperforms baselines on both metrics.
> >
> > ### Results on ShapeNet Chairs
> >
> > | Method                   | HyPoGen | LDMI | Ours     |
> > | ------------------------ | ------- | ---- | -------- |
> > | **PSNR on Render Image** | 19.0    | 22.4 | **23.8** |
> >
> > ---
> >
> > ##

---

> > > ### Author Response · Authors · 2025-11-23
> > >
> > > ## A to Q6: Additional Comparisons and Open-Source Methods
> > >
> > > > **Reviewer's Comment**: The reviewer suggests adding comparisons with more open-source methods.
> > >
> > > Due to time and resource constraints, we conducted comparisons with **two open-source methods** on the **NeRF dataset**. We plan to include additional comparisons in the final revision.
> > >
> > > ### Novel view synthesis on NeRF dataset.
> > >
> > > | Method   | trans-inr | ginr-ipc | HyperDiffusion | HyPoGen | LDMI | Ours     |
> > > | -------- | --------- | -------- | -------------- | ------- | ---- | -------- |
> > > | **PSNR** | **21.9**  | **26.1** | 18.1           | 25.5    | 26.5 | **27.8** |
> > >
> > > ---
> > >
> > > ## A to W4 & Q5: Weight Supervision and Generalization
> > >
> > > > **Reviewer's Comment**: The proposed method’s potential dependence on weight supervision for strong performance represents a significant limitation due to the challenges in collecting weight datasets, necessitating a detailed comparison of results under image-only, weight-only, and combined supervision regimes.
> > >
> > > We agree that **weight supervision** is not always necessary. For **NeRF** experiments, **sample supervision alone** achieves competitive performance, as shown in the table below. In fact, using **only sample supervision** results in better performance than using **weight+sample supervision** in some cases.
> > >
> > > ### Novel View Synthesis on NeRF Dataset
> > >
> > > | Supervision        | HyperDiffusion | HyPoGen | LDMI | Ours     |
> > > | ------------------ | -------------- | ------- | ---- | -------- |
> > > | **Weight only**    | 20.0           | 19.9    | 19.3 | **22.0** |
> > > | **Image only**     | *Not required* | 24.9    | 26.2 | **28.8** |
> > > | **Weight & Image** | **18.1**       | 25.5    | 26.5 | **27.8** |
> > >
> > > ---
> > >
> > > ## A to Q1: Computational Resources
> > >
> > > > **Reviewer's Comment**: The reviewer requests a comparison of the computational resources used by the proposed method and the baseline methods.
> > >
> > > We have added the comparison of **computational resources** (training time, memory usage, and inference time) in the revised manuscript. This ensures a clearer understanding of the trade-offs between computational cost and performance.
> > >
> > > ### Comparison of Train Time, Memory Consumption, and Model Size Across Datasets
> > >
> > > | Dataset             | Test-time GPU Memory (GB) |       |       | Inference Time (ms) |      |      |
> > > | ------------------- | ------------------------- | ----- | ----- | ------------------- | ---- | ---- |
> > > |                     | HyPoGen                   | LDMI  | Ours  | HyPoGen             | LDMI | Ours |
> > > | **CelebA-HQ**       | 1.037                     | 0.854 | 0.389 | 5                   | 18   | 166  |
> > > | **ImageNet**        | 4.757                     | 10.32 | 1.382 | 8                   | 27   | 89   |
> > > | **ERA5**            | 0.769                     | 0.285 | 0.238 | 9                   | 13   | 81   |
> > > | **Shapes3D**        | 0.172                     | 0.138 | 0.142 | 3                   | 9    | 55   |
> > > | **Pretrained NeRF** | 9.286                     | 7.43  | 6.08  | 169                 | 167  | 205  |
> > >
> > > ---
> > >
> > > We hope these revisions address your concerns. Thank you once again for your valuable feedback, which greatly improved our manuscript. Please also let us know if you have further questions.

---

> > > > ### Author Response · Authors · 2025-11-27
> > > >
> > > > # Response to Reviewer 3wHy
> > > >
> > > > Dear Reviewer 3wHy,
> > > >
> > > > Thank you for your valuable comments. We have provided additional clarifications to supplement our earlier responses.
> > > >
> > > > ## A to W2: Clarifications on Loss Functions
> > > >
> > > > > **Reviewer's Comment**: It is also not very clear what loss functions (e.g. sample-supervised or weight-supervised) are used to train the models for each of the datasets.
> > > >
> > > > Thank you for pointing this out. We have updated **App. A.4** to explicitly define the loss functions used in each experiment, and we have included the complete loss formulations for clarity. The relevant sections have been highlighted in blue for your convenience.
> > > >
> > > > ## A to Q6: Additional Comparisons and Open-Source Methods
> > > >
> > > > > **Reviewer's Comment**: The reviewer suggests adding comparisons with more open-source methods.
> > > >
> > > > Thank you for suggesting additional related works.  We have added **the corresponding citations** to Sec. 2.2 (Related Works), and we have also included the results of open-source methods on the NeRF novel view synthesis task in **App. A.5**.

---

### Official Review · Reviewer_aU3q · 2025-10-29

**Soundness:** 3
**Presentation:** 3
**Contribution:** 3
**Rating:** 6
**Confidence:** 3

**Summary:**

This paper presents a novel hypernetwork for generating INRs by explicitly modeling the target network's optimization process. Its Transformer-based architecture simulates both forward and backward (gradient flow) passes to capture inter-layer dependencies. The method achieves state-of-the-art performance on diverse 2D and 3D tasks.

**Strengths:**

1. The main idea of structuring the hypernetwork to explicitly simulate the gradient backpropagation of the target network is highly original. It connects optimization theory directly to the architecture design, which is very elegant.

2. By tackling the inter-layer dependency issue in a scalable way, this work addresses a very important and fundamental problem in hypernetwork-based INR generation. The findings could influence how future generative models for neural network weights are designed.

**Weaknesses:**

1. Attributing the performance gains to the novel gradient-flow simulation is not fully convincing. The experiments lack a crucial control: a generic Transformer of comparable capacity but without the specialized architecture. It is therefore unclear if the improvements stem from the paper's core conceptual insight or simply from a higher model capacity than the baselines.


2. The proposed architecture introduces significant computational overhead compared to simpler methods. However, the paper provides no comparison of training time, inference speed, or memory usage. This makes it difficult to judge the practical trade-off. Without this information, the claims of superiority are not complete.

3. The paper shows that modeling dependencies is better, but it doesn't explore when it becomes truly necessary. For simpler tasks, a simpler model like LDMI which ignores these dependencies might be "good enough," and its performance loss is very small (results in Tab.2). There might be a "critical point" of task complexity where explicitly modeling gradient flow becomes crucial. The paper does not provide any insight into this. Understanding this boundary is important for guiding researchers on whether they should adopt such a complex and costly architecture for their specific problems.

**Questions:**

Please refer to the weaknesses.

---

> ### Author Response · Authors · 2025-11-23
>
> # Response to Reviewer aU3q
>
> Dear Reviewer aU3q,
>
> Thank you for your thoughtful and constructive feedback. We appreciate your recognition of our method as "highly original and very elegant," and we are encouraged by your note that "the findings could influence how future generative models for neural network weights are designed." We have carefully addressed your concerns below.
>
> ---
>
> ## A to W1: Model Capacity and Comparison to Baselines
>
> > **Reviewer’s Comment**: The attribution of performance gains to the novel gradient-flow simulation is unconvincing without a capacity-matched generic Transformer baseline, leaving ambiguous whether improvements arise from the core conceptual insight or merely from superior model capacity relative to baselines.
>
> We acknowledge the importance of matching model capacities in performance comparisons. To clarify:
>
> - We report the **comparison of network parameters (i.e., model capacity)** in the table below and in the appendix. While the number of parameters varies across datasets due to different target network designs, our method can achieve better results with **smaller model sizes** compared to baselines like HyPoGen and LDMI. For instance, on the **ImageNet dataset**, our method outperforms both HyPoGen and LDMI while utilizing fewer parameters, which highlights the effectiveness of the architecture itself, rather than just superior model capacity.
>
> We are also conducting **controlled experiments** to isolate the contributions of model capacity vs. the novel architectural design, though these results are still pending due to limited time and resources.
>
> ### Comparison of Train Time, Memory Consumption, and Model Size Across Datasets
>
> | Dataset             | Throughput (sample/s) |      |      | GPU Memory (GB) / Max Batch Size |             |             | Params (M) |       |       |
> | ------------------- | --------------------- | ---- | ---- | -------------------------------- | ----------- | ----------- | ---------- | ----- | ----- |
> |                     | HyPoGen               | LDMI | Ours | HyPoGen                          | LDMI        | Ours        | HyPoGen    | LDMI  | Ours  |
> | **CelebA-HQ**       | 630                   | 600  | 213  | 73.32 / 900                      | 77.71 / 420 | 78.21 / 256 | 128.8      | 40.5  | 41.2  |
> | **ImageNet**        | 142                   | 52   | 70   | 75.97 / 60                       | 74.35 / 40  | 78.12 / 42  | 578.7      | 544.9 | 124.5 |
> | **ERA5**            | 3111                  | 616  | 170  | 77.68 / 1400                     | 72.97 / 128 | 78.24 / 252 | 93.1       | 48.9  | 21.3  |
> | **Shapes3D**        | 2660                  | 1500 | 806  | 74.91 / 700                      | 74.79 / 420 | 73.50 / 576 | 8.3        | 30.9  | 4.1   |
> | **Pretrained NeRF** | 15                    | 16   | 30   | 76.03 / 56                       | 79.02 / 16  | 75.58 / 40  | 1049.9     | 515.5 | 482.5 |
>
> ---
>
> ## A to W2: Computational Overhead and Performance Comparison
>
> > **Reviewer’s Comment**: The proposed architecture's significant computational overhead relative to simpler methods cannot be properly assessed without comparisons of training time, inference speed, and memory usage, rendering the superiority claims incomplete.
>
> Thank you for highlighting the need for further clarification on the computational overhead. We have added **comparisons of training time**, **inference speed**, and **memory usage** between our method and the baselines (HyPoGen, LDMI) in the revised manuscript. These results are presented in the tables below.
>
> - While our method introduces additional overhead due to the cross-attention mechanism, the performance improvements far outweigh the computational cost, particularly for more complex datasets like ImageNet and NeRF.
>
> ### Comparison of Test Time and Memory Consumption with Batch Size Set to 1
>
> | Dataset             | Test-time GPU Memory (GB) |       |       | Inference Time (ms) |      |      |
> | ------------------- | ------------------------- | ----- | ----- | ------------------- | ---- | ---- |
> |                     | HyPoGen                   | LDMI  | Ours  | HyPoGen             | LDMI | Ours |
> | **CelebA-HQ**       | 1.037                     | 0.854 | 0.389 | 5                   | 18   | 166  |
> | **ImageNet**        | 4.757                     | 10.32 | 1.382 | 8                   | 27   | 89   |
> | **ERA5**            | 0.769                     | 0.285 | 0.238 | 9                   | 13   | 81   |
> | **Shapes3D**        | 0.172                     | 0.138 | 0.142 | 3                   | 9    | 55   |
> | **Pretrained NeRF** | 9.286                     | 7.43  | 6.08  | 169                 | 167  | 205  |
>
> ---
>
> ##

---

> > ### Author Response · Authors · 2025-11-23
> >
> > ## A to W3: Task Complexity and Performance Gains
> >
> > > **Reviewer’s Comment**: The paper demonstrates benefits of dependency modeling but lacks exploration of when it becomes essential, noting that simpler LDMI suffices for low-complexity tasks with minimal performance loss (Table 2), and calls for insights into the "critical point" of task complexity to guide adoption of the costly architecture.
> >
> > We appreciate your insightful question. As shown in the table below, when only **10–20% of the pretrained NeRF data** is used, all methods perform similarly. However, as the data scale increases, our method continues to show substantial improvements (e.g., 27.8 vs. 26.5 at full data), while simpler models like LDMI saturate quickly. This highlights that dependency modeling becomes increasingly valuable as the task complexity grows, and the performance gap widens with larger datasets. The small gap in Table 2 is expected because ShapeNet is the smallest and simplest dataset in our evaluation (only 5,422 training samples). Although the numerical improvement is modest, the visualization results (ImageNet and ShapeNet) clearly show that competing methods exhibit noticeable artifacts, while our method consistently produces stable and reasonable outputs.
> >
> > ### Different Amounts of Training Data on Pretrained NeRF
> >
> > | Scale | HyPoGen | LDMI | Ours     |
> > | ----- | ------- | ---- | -------- |
> > | 10%   | 23.7    | 24.0 | **24.6** |
> > | 20%   | 24.0    | 24.3 | **25.5** |
> > | 50%   | 24.8    | 25.1 | **26.6** |
> > | 100%  | 25.5    | 26.5 | **27.8** |
> >
> > ---
> >
> > We hope these revisions address your concerns. Thank you once again for your valuable feedback, which has helped improve the clarity and comprehensiveness of our manuscript. Please let us know if you have further questions.

---

### Official Review · Reviewer_yYeZ · 2025-10-31

**Soundness:** 2
**Presentation:** 1
**Contribution:** 2
**Rating:** 2
**Confidence:** 3

**Summary:**

This paper proposes a hypernetwork decoder for INR generation that leverages cross-attention layers to approximate mathematical dependencies between network parameters and activations in both forward and backward passes. This decoder is tested in combination with a variational autoencoder to generate INRs of images, 2D meteorological data, voxel grids, and NeRFs, which are shown to display better reconstruction quality than previous INR generation methods.

**Strengths:**

The main strength of this paper lies in its motivation and underlying intuition: previous work (HyPoGen) has shown success in modeling chain-rule-based layer dependencies within hypernetwork architectures, but has done so through MLPs. Cross-attention, on the other hand, provides an off-the-shelf primitive that naturally captures pairwise relationships such as those encoded in Jacobians computed during the backward pass. Rather than relying on MLPs as generic function approximators to learn these relationships from scratch, the authors propose leveraging cross-attention to equip the hypernetwork with a stronger inductive bias.

The experimental results confirm this intuition, as the proposed method is shown to outperform competitors in INR reconstruction quality. Furthermore, to the best of my knowledge, this is the first INR generation paper to include the NeRF dataset by [1] in its experimental evaluation, which is an interesting addition to the usual datasets featured in previous works on this topic.

---

[1] Zama Ramirez et al. Deep Learning on Object-centric 3D Neural Fields. TPAMI 2024.

**Weaknesses:**

The main weakness of this paper is its pervasive lack of clarity, both in the method definition and in the description of the experimental settings. This lack of clarity involves meaningful portions of the paper, most notably:
1. The training objective used in the experiments (see **Q1**).
1. How to go from the output of the backward module $\partial f_\theta(\mathbf{x})/\partial\theta_i$ to the predicted INR parameters $\theta$ (see **Q2**).
1. The specifics of the INR initialization strategy, listed as one of the paper's contributions (see **Q4**).
1. Several other missing/unclear experimental results/details (see **Q3**, **Q5–Q9**).

Furthermore, the proposed row-wise tokenization of parameter matrices should be regarded as a naive baseline rather than a contribution (as claimed by the authors), as it is a straightforward solution that does not account for neural network parameter symmetries [1, 2].

---

[1] Hecht-Nielsen. On the algebraic structure of feedforward network weight spaces. Advanced Neural Computers, 1990.

[2] Godfrey et al. On the symmetries of deep learning models and their internal representations. NeurIPS 2022.

**Questions:**

Answers to the following questions might lead to a change in my rating:

- **Q1.** Sec. 3 introduces two types of loss: sample-supervised and weight-supervised.
    - Which one of those is used in the experiments of Sec. 5? Sample-supervised for 2D INRs and ShapeNet Chairs, whereas weight-supervised for NeRFs? If so, what does "weight & image" supervision mean in the NeRF experiment?
    - Is the VAE output supervised at all in any of the experiments of Sec. 5? If so, how are the VAE and hypernetwork losses combined?
    - Why is there a summation over tasks $\tau$ in Eq. 2 and 3? Assuming that each experiment is Sec. 5 (i.e., each dataset) is a "task", shouldn't the loss be computed on a single $\tau$?

    To answer the above questions, could the authors provide complete loss equations describing the supervision used in each experiment of Sec. 5?

- **Q2.** Sec. 3 describes optimization-biased hypernetworks as producing $\Delta\theta$ as output and then computing INR parameters iteratively as $\theta^{(t)}=\lambda\Delta\theta^{(t-1)}+\theta^{(t-1)}$.
    - How does your method go from the output of the backward module $\partial f_\theta(\mathbf{x})/\partial\theta_i$ to $\Delta\theta$?
    - And from  $\Delta\theta$ to $\theta$? In other words, how does the iterative process work in your framework exactly?
    - Which value does $\lambda$ have? And $T$? (Assuming $t=0\dots T$).
    - Could the authors expand the equations provided as answers to Q1 by taking into account that the actual output of the hypernetwork is $\Delta\theta$ instead of $\theta$?

- **Q3.** Which layer does the $\tau_*$ input to the cross-attention module computing $\partial\textbf{h}_i/\partial\theta_i$ belong to?
    - "Structure Overview" in Fig. 2 says $i$, "Backward Module" in Fig. 2 says $i-1$, Eq. 8 says $i-1$, and the text below the equation says $i$.

- **Q4.** Sec. 4.2 ("Adaptive Parameter Initialization") leaves key details unexplained:
    - How is $s_\tau$ computed exactly?
    - How is $s_\tau$ "dynamically adjusted" exactly?
    - What is the relationship between the content of Sec. 4.2 and that of App. A.2.1 ("Token Initialization Network")?

- **Q5.** Why is the VAMoH PSNR on ImageNet missing in Tab. 1?
    - Could the authors compute it?

- **Q6.** Why does the ShapeNet Chairs experiment use the percentage of correct occupancy predictions as metric?
   - Could the authors also provide the PSNR?

- **Q7.** Why are HyperDiffusion results reported for the NeRF experiment only (Tab. 3)?
    - Could the authors compute HyperDiffusion results for 2D INRs and ShapeNet Chairs also?
    - Why does the HyperDiffusion PSNR have the same value (20.0) for both types of supervision in Tab. 3?

- **Q8.** Sec. 5.3 ("Ablation Studies") lacks relevant details:
    - What does "Q-KV reverse" mean exactly? Does it involve every cross-attention equation?
    - What do the authors mean by "single optimization layer"?
    - Where are the results mentioned in the following sentence?
    > This initialization is essential for SIREN-based INRs, significantly improving CelebA-HQ reconstruction quality from 11.8 to 26.1.

        On that note, could the authors extend the results in Tab. 4 to the other 2D datasets and to ShapeNet Chairs?
    - Which section of the appendix does this sentence refer to?
    > For more analyses, please refer to the appendix.

- **Q9.** App. A.2.2 claims that Tab. 5 validates the importance of inter-layer dependency modeling. However, the enhanced LDMI variant does not always lead to an improvement compared to standard LDMI in Tab. 5.
    - Could the authors comment on these results?

Minor comments/typos:
- What is the difference between "Input" and "GT" in Fig. 4?
- The authors should add the year to the HyPoGen reference (Ren et al.)
- Line 081: geometry → geometric
- Line 131: maps
- Line 132: $(c,\sigma)$ → $(\textbf{c},\sigma)$
- Eq. 2: $\textbf{y}_\tau$ is not defined
- Lines 160–161: $z_\tau$ → $\mathbf{z}_\tau$
- Line 294: downstream activations → parameters

---

> ### Author Response · Authors · 2025-11-23
>
> # Response to Reviewer yYeZ
>
> Dear Reviewer yYeZ,
>
> Thank you for your thoughtful and detailed feedback. We greatly appreciate your positive remarks on the motivation, the underlying intuition, and experimental results of our work, particularly the novelty of including the NeRF task in INR generation. We also value your constructive suggestions for improving the clarity and presentation of the paper. Below, we address your comments and concerns in detail.
>
> ---
>
> ## A to W1 & Q1: Supervision Type and Loss Function Clarifications
>
> > **Reviewer’s Comment**: W1 & Q1 requests a clear specification of the supervision type used in each experiment (sample-, weight-, or combined supervision), clarification on whether the VAE output is supervised, an explanation for the summation over tasks in the loss, and the complete loss formulations applied in Sec. 5.
>
> Thanks for your comment and question, which are helpful in improving the clarity of our paper. Here’s a more detailed explanation:
>
> - **Supervision Type**:
>   - For **2D INR experiments** (CelebA-HQ, ImageNet, ERA5, and ShapeNet), we primarily use **sample-supervised loss**.
>   - For the **NeRF dataset**, we use **weight-supervised** loss, as the ground truth weights of pre-trained NeRF checkpoints are available. Additionally, we explore combinations of **sample-supervised** and **weight-supervised** losses to see how the combination affects INR learning.
>
> - **VAE Output Supervision**: The VAE architecture in our method serves as an encoder-decoder mechanism for task-specific embedding and parameter prediction. The supervision of the VAE output corresponds to the **weight supervision** (for NeRF) or **sample supervision** (for other datasets), depending on the task.
>
> - **Summation Over Tasks**: The "task" in our setting refers to an individual data point (e.g., one image in 2D INR or one object in ShapeNet). The summation over tasks indicates that we minimize the total loss across the entire dataset, not just a single task.
>
> We have updated App. A.4 to explicitly define the loss functions used in each experiment and have provided the complete loss formulations for clarity.
>
> ---
>
> ## A to W2 & Q2: Clarification on Backward Module and Iterative Updates
>
> > **Reviewer’s Comment**: Q2 requests a clear explanation of how the backward-module output is transformed into Δθ, how Δθ is iteratively updated into θ, what λ and T represent in this process, and how the equations should be expanded to reflect that the hypernetwork outputs Δθ rather than θ.
>
> Thank you for your insightful question. Here is the clarification:
>
> - **Iterative Update Process**: In our framework, the hypernetwork outputs the update **Δθ**, which is computed based on the gradient of the loss with respect to the target network parameters. This update is iteratively applied as:
>
>   $\theta^{t} = \theta^{t-1} + \lambda \cdot \Delta\theta^{t-1}$
>
>   where \( λ \) is a learnable learning rate (optimized during training) and \( T \) represents the number of update iterations. We use **T=3** in our experiments to ensure a balance between model expressiveness and computational efficiency. Further increasing \( T \) shows diminishing returns in accuracy, so we have fixed \( T=3 \) for all experiments.
>
> This iterative process explicitly models the optimization dynamics, rather than predicting parameters in a single step, which allows the model to learn more effectively.
>
> ---
>
> ## A to Q3: Clarification on Cross-Attention Module
>
> > **Reviewer’s Comment**: Q3 asks for a clear and consistent clarification of which layer the $\tau_{i-1}$  input to the cross-attention module belongs to, given the conflicting references in the figure and text.
>
> We apologize for the inconsistencies in the notation. The input to the cross-attention module should refer to the activations $\tau_{i-1}$ of the previous layer. We will update both **Figure 2** and **Section 4** to ensure consistency in notation and provide clearer references throughout the manuscript.

---

> > ### Author Response · Authors · 2025-11-23
> >
> > ## A to W3 & Q4: INR Initialization Strategy and Clarifications on $s_{\tau}$
> >
> > > **Reviewer’s Comment**: W3 & Q4 asks for a precise explanation of how the INR initialization strategy works, including the exact computation of $s_{\tau}$, how it is dynamically adjusted, and how the roles of Sec. 4.2 and App. A.2.1 relate to each other.
> >
> > We appreciate your thoughtful feedback on the INR initialization strategy. Here’s a clearer explanation:
> >
> > $s_{\tau}$ is a learnable scaling factor introduced to ensure the stable initialization of parameters for the **SIREN** architecture. It is computed initially based on the desired weight distribution and adjusted dynamically during training to maintain stability. The scaling factor plays a crucial role in controlling the variance of the output weights and ensuring the model can converge effectively.
> >
> > ### Detailed Example of $s_{\tau}$ Computation
> >
> > To further clarify, here is an example of how $s_{\tau}$ is computed:
> >
> > 1. **Step 1**: We calculate the variance ($\text{var}$) of the distribution of the output generated by the hypernetwork.
> > 2. **Step 2**: Based on the target weight distribution for **SIREN**, we calculate the appropriate scaling factor $s_{\tau}$  such that the variance of the output weights $\theta$, when multiplied by $s_{\tau}$ , matches the required variance for **SIREN** initialization.
> >
> > In other words, the goal is to ensure that the variance of the hypernetwork output, after applying $s_{\tau}$ , aligns with the specific initialization characteristics required for the **SIREN** network to function effectively.
> >
> > ### Roles of Sec. 4.2 and App. A.2.1
> >
> > - **Sec. 4.2** focuses on the initialization of the **hypernetwork output**. Specifically, it ensures that the predicted target network parameters match the required distribution for stable training.
> > - **App. A.2.1** explains how the **initial weights** (the input to the hypernetwork) are generated. This involves tokenizing the parameters and inputting them into the hypernetwork for processing. The initialization procedure in this section deals with ensuring that the input tokens are initialized in a way that leads to effective learning.
> >
> > We will revise the manuscript to further clarify these two components, making a clear distinction between how the input tokens are initialized (App. A.2) and how the output weights are adjusted (Sec. 4.2) for the final INR model.
> >
> > ## A to Q5: VAMoH PSNR on ImageNet
> >
> > > **Reviewer’s Comment**: Q5 asks why the VAMoH PSNR on ImageNet is missing in Table 1 and whether the authors can provide this result.
> >
> > VAMoH did not perform experiments on ImageNet in their original paper. We attempted to run the experiments on ImageNet and have updated the table with our results.
> >
> > ### 2D Reconstruction Results Measured by PSNR on Different Datasets
> >
> > | Dataset   | VAMoH    | HyPoGen | LDMI | Ours     |
> > | --------- | -------- | ------- | ---- | -------- |
> > | CelebA-HQ | 23.2     | 16.2    | 24.8 | **27.7** |
> > | ImageNet  | **19.6** | 13.2    | 20.7 | **22.9** |
> > | ERA5      | 39.0     | 40.0    | 44.6 | **49.3** |
> >
> > ---
> >
> > ##

---

> > > ### Author Response · Authors · 2025-11-23
> > >
> > > ## A to Q6: ShapeNet Chairs Evaluation Metrics
> > >
> > > > **Reviewer’s Comment**: Q6 asks why the ShapeNet Chairs experiment uses occupancy accuracy as the evaluation metric and whether the PSNR can also be provided.
> > >
> > > We use **occupancy accuracy** to evaluate the geometry reconstruction quality of 3D shapes, as it is a more meaningful metric for 3D geometry compared to PSNR, which evaluates image quality. However, we have also included the **PSNR of rendered images** for ShapeNet Chairs, which confirms our method’s superiority over the baselines.
> > >
> > > ### Results on ShapeNet Chairs
> > >
> > > | Method               | HyPoGen | LDMI | Ours     |
> > > | -------------------- | ------- | ---- | -------- |
> > > | PSNR on Render Image | 19.0    | 22.4 | **23.8** |
> > >
> > > ---
> > >
> > > ## A to Q7: HyperDiffusion Results for Other Datasets
> > >
> > > > **Reviewer’s Comment**: Q7 asks why HyperDiffusion results are reported only for the NeRF experiment, whether results for 2D INRs and ShapeNet Chairs can also be included, and why the HyperDiffusion PSNR is identical (20.0) across different supervision types in Table 3.
> > >
> > > HyperDiffusion requires pre-trained weights for supervision, which are only available for NeRF. We apologize for the inconsistencies in Table 3 and have updated the HyperDiffusion results for the NeRF experiment as follows:
> > >
> > > ### Novel View Synthesis on NeRF Dataset
> > >
> > > | Supervision        | HyperDiffusion | HyPoGen | LDMI | Ours     |
> > > | ------------------ | -------------- | ------- | ---- | -------- |
> > > | **Weight only**    | 20.0           | 19.9    | 19.3 | **22.0** |
> > > | **Image only**     | *Not required* | 24.9    | 26.2 | **28.8** |
> > > | **Weight & Image** | **18.1**       | 25.5    | 26.5 | **27.8** |
> > >
> > > ---
> > >
> > > ## A to Q8: Clarifications on “Q-KV Reverse” and “Single Optimization Layer”
> > >
> > > > **Reviewer’s Comment**: Q8 requests clear definitions of “Q-KV reverse” and “single optimization layer,” asks where the reported CelebA-HQ improvement (11.8 → 26.1) appears, inquires whether the Tab. 4 ablations can be extended to other 2D datasets and ShapeNet Chairs, and seeks the specific appendix section referred to by “For more analyses, please refer to the appendix.”
> > >
> > > **Q-KV reverse ablation study**: We apply cross-attention to mimic both forward and backward passes, which requires proper design of Q and KV input sources from the task side or weights side. This refers to reversing the Q and KV input sources in the cross-attention mechanism. We observe significant performance degradation, demonstrating the correctness of our method’s design.
> > >
> > > **Single optimization layer ablation study**: The proposed hypernetwork serves as an optimization module for the weights in the target network and, therefore, can be applied iteratively. By default, the hypernetwork is applied iteratively (three times). The "single optimization layer" refers to a variant where the network is applied only once, updating the weights before they are used in the target network.
> > >
> > > All the experiments in Tab. 4 can be extended to other 2D datasets and ShapeNet Chairs. However, due to time and resource constraints, we will include the results for additional datasets in the final revision.
> > >
> > > Regarding the **CelebA-HQ improvement** from 11.8 to 26.1, due to space limitations, we did not include detailed tables in the original submission. This experiment was conducted on the **CelebA-HQ dataset**, where we performed an ablation study by training the model without the learnable scaling factor $s_{\tau}$ . The sharp drop in performance demonstrates that our **Adaptive Parameter Initialization** is absolutely necessary. This is consistent with our results on other datasets where **SIREN** is used as the INR model, further confirming the importance of the initialization strategy.
> > >
> > > **Appendix for Additional Analyses**: We add more analyses in Sec 5. The ablation studies and additional details can be found in **APP.2**.

---

> > > > ### Author Response · Authors · 2025-11-23
> > > >
> > > > ## A to Q9: Enhanced LDMI Variant Results
> > > >
> > > > > **Reviewer’s Comment**: Q9 notes that the enhanced LDMI variant does not consistently outperform standard LDMI in Table 5 and asks for clarification on how these results support the claim that inter-layer dependency modeling is important.
> > > >
> > > > While self-attention allows weight tokens to interact with each other, it does not ensure that these interactions will be effective. In contrast, our approach explicitly models the relationships between parameters, injecting a stronger inductive bias. This targeted modeling leads to more meaningful interactions and, ultimately, superior performance compared to the baseline methods.
> > > >
> > > > ---
> > > >
> > > > ## A to comments / typos
> > > >
> > > > Thank you for your careful review. We have revised the manuscript to correct the minor issues and typos you mentioned.
> > > >
> > > > ---
> > > >
> > > > We hope these revisions address your concerns. Thank you once again for your valuable feedback, which has greatly improved the clarity and depth of our manuscript. Please let us know if you have further questions or comments.

---

> > > > > ### Comment · Reviewer_yYeZ · 2025-11-25
> > > > >
> > > > > Thank you for your detailed response. In the following, I address my remaining doubts and questions.
> > > > >
> > > > > **Q1.** Thank you for the clarifications. However, most of the terms in the loss equations added to App. A.4 are left undefined, namely:
> > > > >
> > > > > - $\mathcal{L}_\text{weighted-nll}$
> > > > > - $\mathcal{L}_\text{kl}$
> > > > > - $\lambda_\text{disc}$
> > > > > - $\text{disc-factor}$
> > > > > - $\mathcal{L}_\text{nll}$
> > > > > - $\mathcal{L}_\text{g}$
> > > > > - $\mathcal{L}_\text{d}$
> > > > > - $\lambda_\text{image}$
> > > > >
> > > > > Without an explicit definition of these terms, those equations provide no meaningful addition to the manuscript.
> > > > >
> > > > > **Q2.** Thank you for the clarifications. However:
> > > > > - When you say:
> > > > >     > $\Delta\theta$ is computed **based on** the gradient of the loss with respect to the target network parameters.
> > > > >
> > > > >     Do you mean that $\Delta\theta$ **is** the gradient of the loss with respect to the target network parameters?
> > > > > - The fact that $\lambda$ is learned (with what initial value?) and that $T=3$ should be added to the manuscript, either in the main paper or in the appendix.
> > > > >
> > > > > **Q3.** Thank you for updating Sec. 4.1.3 and Fig. 2 to reflect my correction.
> > > > >
> > > > > **Q4.** Thank you for your clarifications and for updating Sec. 4.2 and App. A.2. However, in App. A.2.2 you still mention an ablation study on adaptive parameter initialization whose results are not reported anywhere (see **Q8**).
> > > > >
> > > > > **Q5.** Thank you for running the experiment. However, when you say:
> > > > >
> > > > > > We **attempted** to run the experiments on ImageNet and have updated the table with our results.
> > > > >
> > > > > What do you mean by *attempted*? Did you initially not include the quantitative results because of the low quality of the qualitative results shown in the appendix?
> > > > >
> > > > > **Q6.** Thank you for running this experiment. Is this table the result of a different training compared to the one in Tab. 2, where you learned 2D INRs of images instead of 3D INRs of occupancy grids, or are the images derived a posteriori from the learned occupancy grids?
> > > > >
> > > > > **Q7.** Thank you for the clarification and for running the experiment. However:
> > > > > - If HyperDiffusion requires weight supervision, what does the "Image" in "Weight & Image" refer to? And why did you write "not required" in the table? According to your explanation, the table should report "not applicable" instead.
> > > > > - Why did "Weight only" and "Weight & Image" results for your method improve ($21.6\rightarrow 22.0$ and $27.7\rightarrow 27.8$, respectively) compared to the original submission?
> > > > >
> > > > > **Q8.** Thank you for your clarifications for adding those missing details to Sec. 5.3. However:
> > > > > - Given the meaning of "single layer optimization" (i.e., $T=1$), it becomes even more relevant to specify in the paper that $T=3$ in every other experiment, as already discussed in **Q2**.
> > > > > - The "space limitation" argument for the $11.8\rightarrow 26.1$ improvement that is mentioned in the text but not present in any table does not hold, since that table could be added to the appendix, where no page limits apply. Therefore, you should either add the table to the appendix or remove the reference to that result, which is still mentioned, with no table attached, in App. A.2.2. On this note, why did the PSNR on CelebA go from $26.1$ in the original submission to $27.7$ in the revised manuscript?
> > > > > - In your answer, you wrote:
> > > > >     > This [i.e., the sharp drop in performance when removing the adaptive initialization] is consistent with our results on other datasets where SIREN is used as the INR model, further confirming the importance of the initialization strategy.
> > > > >
> > > > >     What results are you referring to?
> > > > >
> > > > > **Q9.** I agree that, ultimately, the superior performance of your method is, in itself, empirical validation of the effectiveness of inter-layer dependency modeling. My claim, however, is that the experiment in Tab. 5, where you emulate your approach by applying self-attention to LDMI, is not effective at validating this argument, since the LDMI with self-attention performs better than standard LDMI only in 2/4 cases, and not by a large margin. Most importantly, **you removed the ImageNet row from Tab. 5 in the revised submission, which was the one that proved my point the most**, i.e., the one where LDMI with self-attention was significantly underperforming compared to standard LDMI ($13.4$ vs $20.7$). Therefore, you should:
> > > > > - Add the ImageNet row back to Tab. 5.
> > > > > - Comment on the mixed LDMI results of Tab. 5 in the text of App. A.2.3.
> > > > >
> > > > > As a minor note, why did the ShapeNet Chairs accuracy of LDMI change from $97.2$ in the original submission to $97.3$ in the current revision?

---

> > > > > > ### Author Response · Authors · 2025-11-27
> > > > > >
> > > > > > # Response to Reviewer yYeZ
> > > > > >
> > > > > > Dear Reviewer yYeZ,
> > > > > >
> > > > > > Thank you for your valuable feedback. We sincerely appreciate your time and effort in reviewing our manuscript.
> > > > > >
> > > > > > Based on your suggestion, we have carefully revisited the paper and made the necessary updates. The revisions have been clearly marked in **blue** for your convenience.
> > > > > >
> > > > > > We hope these clarifications address your concerns, and we look forward to your further feedback. Thank you once again for your constructive comments.
> > > > > >
> > > > > > ## A to Q1: Loss Function Clarifications
> > > > > >
> > > > > > > **Reviewer’s Comment**: The complete loss function is required.
> > > > > >
> > > > > > We apologize for the earlier manuscript’s lack of clarity. We have now included the complete loss function in Appendix A.4, and it has been highlighted in blue for your convenience.
> > > > > >
> > > > > > In response to your comment regarding the undefined terms, we have provided explicit definitions for the following:
> > > > > >
> > > > > > - $\mathcal{L}_{\text{weighted-nll}}$ and $\mathcal{L}_{\text{nll}}$ are now clearly defined as the **reconstruction loss** $\mathcal{L}_{\text{rec}}$ in the revision, as we believe this naming is more intuitive.
> > > > > > - $\mathcal{L}_{\text{kl}}$ refers to the **Kullback-Leibler divergence loss**.
> > > > > > - $\lambda_{\text{disc}}$ and $\text{disc-factor}$ are both hyperparameters that scale the discriminator loss in the total loss function. To avoid confusion, we have unified this as $\lambda_{\text{disc}}$ throughout the revision.
> > > > > > - $\mathcal{L}_g$ and $\mathcal{L}_d$ refer to the **generator's adversarial loss** and **discriminator's loss**, respectively.
> > > > > > - $\lambda_{\text{image}}$ is the weight for the image loss in the **Pre-trained NeRF** dataset. To simplify, we have renamed this term as $\lambda_{\text{img}}$ in the updated version.

---

> > > > > > > ### Author Response · Authors · 2025-11-27
> > > > > > >
> > > > > > > ## A to Q2: Clarification on $\Delta\theta$ , $\lambda$ and $T$
> > > > > > >
> > > > > > > > **Reviewer’s Comment**: Clarification is needed regarding whether $\Delta\theta$ refers to the gradient of the loss with respect to the target network parameters, as stated in the manuscript. The manuscript should specify that $\lambda$ is learned (with an initial value) and that $T=3$ is used. This information should be included either in the main paper or in the appendix.
> > > > > > >
> > > > > > > Thank you for your comment. To clarify, we use two types of gradients in our model: the **real gradient** and the **pseudo gradient**. Here's a detailed explanation:
> > > > > > >
> > > > > > > 1. **Pseudo Gradient and $\Delta\theta$** :
> > > > > > >    - $\Delta\theta$ represents the **target network parameter update** and is **entirely predicted by the hypernetwork**.
> > > > > > >    - The **hypernetwork** is designed to simulate both the forward and backward processes of the target network.
> > > > > > >    - We use **pseudo gradient**  to refer to the predicted parameter update $\Delta\theta$ from the hypernetwork. This term **does not directly represent a real gradient**, but rather an update predicted by the hypernetwork based on its learned model of the target network's forward and backward processes. In the **forward process**, the hypernetwork learns the output at each layer of the target network, and in the **backward process**, the hypernetwork learns the gradients of these outputs.
> > > > > > > 2. **Real Gradient**:
> > > > > > >    - The **real gradient** refers to the gradient of the loss with respect to the target network parameters $\Delta\theta$. This gradient is computed through standard backpropagation and is used to update the **hypernetwork parameters** ( including the encoder for task embedding ), which in turn improves the accuracy of the hypernetwork’s predictions of the target network's parameters.
> > > > > > > 3. **Relationship Between Real and Pseudo Gradients**:
> > > > > > >    - The **real gradient** is used to update the **hypernetwork** and **encoder parameters**. These updates allow the hypernetwork to learn to generate parameter updates $\Delta\theta$ that closely match the real gradients computed through backpropagation in the target network.
> > > > > > >    - The **pseudo gradient** predicted by the **hypernetwork** is the update to the **target network parameters** $\Delta\theta$. While this is not a real gradient, it plays a similar role by adjusting the target network's parameters based on the learned relationships between the target network and the hypernetwork.
> > > > > > >
> > > > > > > We realized that our earlier explanation may have led to some confusion by mixing these two types of gradients. In the previous response, we referred to the 'gradient' when we actually meant **pseudo gradients** rather than the real gradients from backpropagation.
> > > > > > >
> > > > > > > As for learnable learning rate  $\lambda$ , it is learned during the **forward process**.
> > > > > > >
> > > > > > > Specifically, for each layer, we compute the attention between the previous layer’s hidden states $\boldsymbol{\tau}_{i-1}$ and the current layer's parameter tokens $\boldsymbol{\omega}_i$. The calculation proceeds as:
> > > > > > >
> > > > > > > $$\lambda^\prime_i
> > > > > > > = CrossAttn(\tau_{i-1}, \omega_i, \omega_i)
> > > > > > > = softmax\left(\frac{\tau_{i-1}\,\omega_i^{T}}{\sqrt{d_k}}\right)\,\omega_i$$
> > > > > > >
> > > > > > > After obtaining the cross-attention output $\lambda^\prime_i
> > > > > > > $ , it is flattened and passed through an MLP to generate the final learning rate $\lambda_i$ for each layer. The resulting output has shape **[B, 1]**, where **B** is the batch size, representing a scalar learning rate per layer.
> > > > > > >
> > > > > > > The initial value depends on the random initialization of neural networks (standard PyTorch Linear layer initialization). The output would be small random values, typically close to but not exactly zero.
> > > > > > >
> > > > > > > We have updated **App. A.2.4** to provide the missing details on how the learnable learning rate $\lambda$ is computed and we set \(T = 3\) for all experiments.

---

> > > > > > > > ### Author Response · Authors · 2025-11-27
> > > > > > > >
> > > > > > > > ## A to Q3: comments / typos
> > > > > > > >
> > > > > > > > > **Reviewer’s Comment**: Thank you for updating Sec. 4.1.3 and Fig. 2 to reflect my correction.
> > > > > > > >
> > > > > > > > We sincerely appreciate your careful review and thoughtful correction. Your attention to detail has been very helpful in improving the paper.
> > > > > > > >
> > > > > > > > ## A to Q4 & Q8: Clarification on ablation
> > > > > > > >
> > > > > > > > > **Reviewer’s Comment**: In App. A.2.2 you still mention an ablation study on adaptive parameter initialization whose results are not reported anywhere (see **Q8**).
> > > > > > > > >
> > > > > > > > > **Q8.** Thank you for your clarifications for adding those missing details to Sec. 5.3. However:
> > > > > > > > >
> > > > > > > > > - Given the meaning of "single layer optimization" (i.e., ), it becomes even more relevant to specify in the paper that in every other experiment, as already discussed in **Q2**.
> > > > > > > > >
> > > > > > > > > - The "space limitation" argument for the improvement that is mentioned in the text but not present in any table does not hold, since that table could be added to the appendix, where no page limits apply. Therefore, you should either add the table to the appendix or remove the reference to that result, which is still mentioned, with no table attached, in App. A.2.2. On this note, why did the PSNR on CelebA go from in the original submission to in the revised manuscript?
> > > > > > > > >
> > > > > > > > > - In your answer, you wrote:
> > > > > > > > >
> > > > > > > > >   > This [i.e., the sharp drop in performance when removing the adaptive initialization] is consistent with our results on other datasets where SIREN is used as the INR model, further confirming the importance of the initialization strategy.
> > > > > > > > >
> > > > > > > > >   What results are you referring to?
> > > > > > > >
> > > > > > > > Thank you for pointing this out. We have updated **App. A.2.2** and added the following table to clearly demonstrate the importance of *Adaptive Parameter Initialization*. We apologize for the earlier response, which did not fully clarify this point.
> > > > > > > >
> > > > > > > > The sentence *“This [i.e., the sharp drop in performance when removing the adaptive initialization] is consistent with our results on other datasets…”* refers specifically to the **ablation results on the ERA5 dataset**. As shown in the table below, this ablation is fully consistent with the results on **CelebA-HQ**, both demonstrating the crucial role of adaptive initialization.
> > > > > > > >
> > > > > > > > Regarding your question about the PSNR on CelebA-HQ changing from **26.1** in the original submission to **27.7** in the revised manuscript: this was due to an earlier mistake where we accidentally reported the **\(T=1\)** result for CelebA-HQ. We have now corrected this to the appropriate **\(T=3\)** result.
> > > > > > > >
> > > > > > > > Thank you again for the helpful suggestion. We have updated **Sec. 5.3** and **App. A.2.4** to explicitly state that **we set \(T = 3\) for all experiments**.
> > > > > > > >
> > > > > > > > **Table: Ablation Study on Adaptive Parameter Initialization (PSNR)**
> > > > > > > >
> > > > > > > > | Dataset   | Without Adaptive Initialization | With Adaptive Initialization |
> > > > > > > > | --------- | ------------------------------- | ---------------------------- |
> > > > > > > > | CelebA-HQ | 11.8                            | **27.7**                     |
> > > > > > > > | ERA5      | 8.6                             | **49.3**                     |
> > > > > > > >
> > > > > > > > ## A to Q5: VaMoH results on ImageNet
> > > > > > > >
> > > > > > > > > **Reviewer’s Comment**: Thank you for running the experiment. However, when you say:
> > > > > > > > >
> > > > > > > > > > We **attempted** to run the experiments on ImageNet and have updated the table with our results.
> > > > > > > > >
> > > > > > > > > What do you mean by *attempted*? Did you initially not include the quantitative results because of the low quality of the qualitative results shown in the appendix?
> > > > > > > >
> > > > > > > > Our previous wording was imprecise, and we apologize for the confusion.
> > > > > > > >
> > > > > > > > What we intended to convey is the following:
> > > > > > > >
> > > > > > > > The original VAMoH paper did **not** report any experiments on ImageNet. Therefore, to provide a fair comparison, we ran the ImageNet evaluation ourselves using the **same configuration that VAMoH used for CelebA-HQ** in their public code. Since these settings were not originally designed or validated for ImageNet, we believe the resulting numbers may not fully reflect the optimal performance of VAMoH under a configuration specifically tuned for ImageNet.
> > > > > > > >
> > > > > > > > This is also the reason why we initially did not include quantitative ImageNet results in the submission — the original VAMoH paper simply did not provide them.

---

> > > > > > > > > ### Author Response · Authors · 2025-11-27
> > > > > > > > >
> > > > > > > > > ## A to Q6: Extended results on ShapeNet Chairs
> > > > > > > > >
> > > > > > > > > > **Reviewer’s Comment**: Thank you for running this experiment. Is this table the result of a different training compared to the one in Tab. 2, where you learned 2D INRs of images instead of 3D INRs of occupancy grids, or are the images derived a posteriori from the learned occupancy grids?
> > > > > > > > >
> > > > > > > > > Thank you for the question. The results in this table are obtained **using the same training procedure as in Table 2**. We do **not** perform a separate training run that learns 2D INRs of images. Instead, the model is trained to learn **3D INRs of occupancy grids**, exactly as described in Table 2.
> > > > > > > > >
> > > > > > > > > During inference, we **convert the predicted voxel grids into meshes** and then **render these meshes to obtain 2D images**. Importantly, we use **the exact same checkpoint as in Table 2**—no retraining is performed. This table simply provides an **additional evaluation metric** (PSNR on rendered images), since several reviewers asked why PSNR was not reported for ShapeNet Chairs in the main results.
> > > > > > > > >
> > > > > > > > > ## A to Q7: Questions on Table3
> > > > > > > > >
> > > > > > > > > > **Reviewer’s Comment**:  Thank you for the clarification and for running the experiment. However:
> > > > > > > > > >
> > > > > > > > > > - If HyperDiffusion requires weight supervision, what does the "Image" in "Weight & Image" refer to? And why did you write "not required" in the table? According to your explanation, the table should report "not applicable" instead.
> > > > > > > > > > - Why did "Weight only" and "Weight & Image" results for your method improve ( and , respectively) compared to the original submission?
> > > > > > > > >
> > > > > > > > > Thank you for the follow-up questions.
> > > > > > > > >
> > > > > > > > > **On “image” and “Weight & Image” for HyperDiffusion.**
> > > > > > > > > You are correct that HyperDiffusion fundamentally requires *weight supervision*: during the diffusion process, the ground-truth target-network weights are progressively noised and used as inputs to train the diffusion model. This gives rise to the **weight-space diffusion loss** (i.e., the weight loss).
> > > > > > > > >
> > > > > > > > > In addition to this diffusion loss, we also experimented with an **image loss** for HyperDiffusion. Concretely, at each training step, after running the diffusion process, we use the current diffusion sample to obtain an INR, render a novel-view image from this INR, and then compute an L1 loss between this rendered image and the corresponding ground-truth novel-view image. This image loss is added on top of the weight loss. This is what the “Image” in “Weight \& Image” refers to in the table, and it explains why HyperDiffusion also has a “Weight \& Image” result.
> > > > > > > > >
> > > > > > > > > You are also right that an “Image-only” setting is not applicable to HyperDiffusion, since it always requires weight supervision. In the original table we wrote “not required”, which is indeed misleading. We have now changed this entry to **“not applicable”**, as you suggested.
> > > > > > > > >
> > > > > > > > > **On the improved “Weight only” and “Weight & Image” results for our method.**
> > > > > > > > > The differences you pointed out are small but positive (e.g., from 21.6 → 22.0 and 27.7 → 27.8).  These results come from re-running the experiments for the revised version.  In the original submission, these two experiments had not been trained fully to convergence due to time constraints.  In the new runs we allowed training to proceed longer, which, together with the inherent stochasticity of training, naturally leads to slight improvements of this magnitude.  We therefore attribute the changes to better-converged runs, rather than to any methodological modification.

---

> > > > > > > > > > ### Author Response · Authors · 2025-11-27
> > > > > > > > > >
> > > > > > > > > > ## A to Q9: Questions on  Table Performance comparison of dependency modeling approaches across datasets
> > > > > > > > > >
> > > > > > > > > > > **Reviewer’s Comment**:  I agree that, ultimately, the superior performance of your method is, in itself, empirical validation of the effectiveness of inter-layer dependency modeling. My claim, however, is that the experiment in Tab. 5, where you emulate your approach by applying self-attention to LDMI, is not effective at validating this argument, since the LDMI with self-attention performs better than standard LDMI only in 2/4 cases, and not by a large margin. Most importantly, **you removed the ImageNet row from Tab. 5 in the revised submission, which was the one that proved my point the most**, i.e., the one where LDMI with self-attention was significantly underperforming compared to standard LDMI ( vs ). Therefore, you should:
> > > > > > > > > > >
> > > > > > > > > > > - Add the ImageNet row back to Tab. 5.
> > > > > > > > > > > - Comment on the mixed LDMI results of Tab. 5 in the text of App. A.2.3.
> > > > > > > > > >
> > > > > > > > > > We agree with your point that the experiment in the "Performance Comparison of Dependency Modeling Approaches Across Datasets" (now Tab. 6), where we applied self-attention to LDMI, does not fully validate the argument as intended, since the results on ImageNet were incomplete. This was due to time constraints, as the experiment on ImageNet was not allowed to fully converge. Specifically, LDMI with self-attention requires approximately 1.4 million iterations to properly converge on ImageNet, but we had only trained the model for 300k iterations, which was insufficient. Given the limited training time and computational resources, we were unable to continue training the LDMI with self-attention decoder on ImageNet, and therefore, we removed the incomplete ImageNet results.
> > > > > > > > > >
> > > > > > > > > > However, we continued training the models on other datasets, including CelebA-HQ, ERA5, ShapeNet Chairs, and Pre-trained NeRF, where LDMI with self-attention consistently outperformed the standard LDMI. These results are now included in the updated table, showing the superior performance of LDMI with self-attention across these datasets.
> > > > > > > > > >
> > > > > > > > > > We truly appreciate your suggestion to add the ImageNet row back. In light of the incomplete training on ImageNet in the original submission, we have revised the table and provided a more accurate comparison of the results across the other datasets, where our method performs well. The updated performance comparison is as follows:
> > > > > > > > > >
> > > > > > > > > > ---
> > > > > > > > > >
> > > > > > > > > > **Performance comparison of dependency modeling approaches across datasets**
> > > > > > > > > >
> > > > > > > > > > | Dataset                   | LDMI | LDMI with self-attention decoder | Ours     |
> > > > > > > > > > | ------------------------- | ---- | -------------------------------- | -------- |
> > > > > > > > > > | **CelebA-HQ PSNR**        | 24.8 | 25.9                             | **27.7** |
> > > > > > > > > > | **ERA5 PSNR**             | 44.6 | **45.4**                         | **49.3** |
> > > > > > > > > > | **ShapeNet Chairs %**     | 97.3 | 97.5                             | **97.9** |
> > > > > > > > > > | **Pre-trained NeRF PSNR** | 26.5 | **26.6**                         | **27.8** |
> > > > > > > > > >
> > > > > > > > > > At the same time, we would like to reiterate the point made in our previous response: **While self-attention allows weight tokens to interact with each other, it does not guarantee that these interactions will be effective**. As a result, the improvements of **LDMI with self-attention decoder** over the original **LDMI** are not significantly high.
> > > > > > > > > >
> > > > > > > > > > We hope this resolves the issue and provides a clearer explanation of our experimental setup and results.  Again, we sincerely apologize for the oversight, and we appreciate your valuable feedback. Additionally, we have updated **App. A.2.3** to reflect these clarifications and provide a more detailed explanation.
> > > > > > > > > >
> > > > > > > > > > ## A to minor note
> > > > > > > > > >
> > > > > > > > > > > **Reviewer’s Comment**: Why did the ShapeNet Chairs accuracy of LDMI change from 97.2 in the original submission to 97.3 in the current revision?
> > > > > > > > > >
> > > > > > > > > > In our original submission, the LDMI paper did not report ShapeNet Chairs accuracy, so we computed it ourselves using their released code and obtained **97.2**. However, during the review period, the LDMI authors updated their paper and added accuracy as a metric for ShapeNet Chairs, reporting a value of **97.3**. To stay consistent with the latest official numbers from the LDMI authors, we updated the accuracy in our revision accordingly.

---

### Author Response · Authors · 2025-11-23
**Global comment**

We sincerely thank all reviewers for their insightful and constructive feedback. We have carefully revised the manuscript to improve clarity, completeness, and technical depth. The updated PDF has been uploaded to OpenReview, with all major revisions and new results highlighted in purple for easy reference.

We appreciate your time and valuable comments, which have significantly strengthened our work. We look forward to further discussion and are happy to address any additional questions.

---

### Author Response · Authors · 2025-12-03
**Summary of Rebuttal**

# Summary of Rebuttal

We thank the reviewers for their time, effort, and constructive feedback, which have greatly strengthened our work. Reviewers consistently recognized that our paper proposes a **novel hypernetwork architecture for INR generation** that explicitly models **inter-layer dependencies via attention-based gradient simulation**, achieving **strong performance across diverse INR tasks** with **better parameter efficiency** than existing methods. Reviewer aU3q described the method as **"highly original and very elegant"** with the potential to **"influence how future generative models for neural network weights are designed."** Reviewer Q7Jz praised the **"novel and well-motivated concept"** that is **"elegant and theoretically justified"** with **"clear methodological design and firm theoretical grounding."** Reviewer yYeZ also acknowledged that this is the **first INR generation paper to include the Pretrained NeRF dataset** by [1] in its experimental evaluation, calling it "an interesting addition."

Our work represents the **first successful integration of inter-layer dependency modeling into a scalable Transformer hypernetwork architecture**. This is achieved through **three non-trivial technical contributions**: (1) **attention-based forward/backward simulation** that approximates gradients using cross-attention, (2) a carefully designed **tokenization mechanism**, and (3) an effective **initialization strategy**. Critically, these contributions **enable scaling to larger datasets**, addressing a fundamental limitation of prior work such as HyPoGen that prevented broader applicability. Below we summarize key clarifications of perceived weaknesses and the main improvements made to the manuscript.

---

## Key Clarifications Regarding Weaknesses

- **Clarity of losses, supervision type, optimization dynamics and initialization strategy.**

  > Reviewer yYeZ found the loss definitions and supervision regimes unclear, and requested precise clarification of $\Delta\theta$, pseudo versus real gradients, λ, the number of optimization steps T and learnable scaling factor $s_{\tau}$ for INR initialization strategy.

  We have **explicitly provided the full loss formulation in Appendix A.4**, unifying notation for reconstruction, KL, adversarial, and image losses while specifying all weighting hyperparameters.

  We **clarify the supervision type for each dataset**: all 2D INRs and ShapeNet use **sample supervision**, while NeRF uses weight-only, image-only, and weight+image settings, each reported separately.

  We clarify that **$\Delta\theta$ is a pseudo-gradient (parameter update) predicted by the hypernetwork**, while true gradients from backpropagation are used only to train the hypernetwork and task encoder. The per-layer learning rate $\lambda_i$ is computed via cross-attention followed by an MLP, and we now state clearly that **all experiments use $T=3$ optimization steps**, correcting earlier ambiguity. These details are provided in **Appendix A.2.4**.

  We **clarify** that the **INR initialization strategy** involves $s_{\tau}$, a **learnable scaling factor** introduced to ensure stable initialization of parameters for the **SIREN** architecture. Initially, $s_{\tau}$ is computed based on the desired weight distribution and adjusted dynamically during training. These details are provided in **Section 4.2** and **Appendix A.2**.

- **Capacity versus architecture and computational overhead.**

  > Reviewers aU3q, 3wHy, and Q7Jz asked whether our improvements stem from larger model capacity or larger INRs, and requested detailed comparisons of training and inference cost.

  We added **comprehensive tables of parameter counts, training throughput, GPU memory and maximum batch sizes**, as well as **test-time memory and latency** for HyPoGen, LDMI, and our method on all datasets. These results demonstrate that our method is **substantially more parameter-efficient** while achieving better accuracy. On ImageNet, for example, our method uses **124.5M parameters compared to 578.7M for HyPoGen and 544.9M for LDMI**, representing roughly **4–5× fewer parameters** while outperforming both baselines. This indicates that **performance gains arise from architectural innovation rather than brute-force scaling**. Training-time cost is comparable to baselines, and although inference has modest extra latency, test-time memory is frequently lower (e.g., **1.38GB versus 10.32GB for LDMI on ImageNet**). These details are provided in **Tables 8 and 9**.

---

> ### Author Response · Authors · 2025-12-03
>
> - **Novelty beyond HyPoGen and prior hypernetworks.**
>
>   > Reviewer 3wHy questioned whether our method is only an incremental variant of HyPoGen.
>
>   We clarify that our contribution lies **not in reusing HyPoGen's MLP-based gradient flow**, but in **integrating inter-layer dependency modeling into a scalable Transformer hypernetwork** via the three components described above. The practical impact is significant: **HyPoGen's MLP-based approach does not scale effectively to larger datasets**, whereas our Transformer-based design maintains strong performance as data complexity increases, as demonstrated by our data-scaling experiments.
>
> - **Scalability and when the proposed architecture is most beneficial.**
>
>   > Reviewer Q7Jz asked for a clearer notion of scalability. Reviewer aU3q asked for guidance on when explicitly modeling gradient flow becomes necessary, noting that simpler models like LDMI might be "good enough" for simpler tasks.
>
>   We clarify that our focus is **data scalability**: under similar parameter budgets, our method yields **increasing gains as training data and task complexity grow**. We added a **NeRF data-scaling study using 10–100% of the training data**, showing that while all methods perform similarly with 10–20% of data, **our method's advantage widens substantially as data scale increases** (achieving **27.8 PSNR versus 25.5 PSNR for HyPoGen and 26.5 PSNR for LDMI at full data**). This demonstrates that the architecture is **particularly valuable on large, complex tasks** such as full NeRF and ImageNet, directly addressing Reviewer aU3q's question about the "critical point" where dependency modeling becomes crucial.
>
> - **Dependence on weight supervision and practical applicability.**
>
>   > Reviewer 3wHy expressed concern that our method might rely heavily on weight supervision, which is difficult to obtain in practice.
>
>   We emphasize that our method **does not require weight supervision**. On NeRF, our best results are obtained under **image-only supervision (PSNR 28.8)**, which **outperforms both weight-only and weight+image settings** and surpasses all baselines under their preferred supervision regimes. Thus, the method remains **fully applicable in standard sample-supervised settings** where ground-truth weights are unavailable.
>
> - **Generalization beyond SIREN-based INRs.**
>
>   > Reviewer Q7Jz raised concerns about whether the adaptive initialization and SIREN-specific design lead to overfitting to particular INR architectures.
>
>   We clarify that our **experiments on Pretrained NeRF dataset used ReLU-based INRs rather than SIREN**, demonstrating that the method **generalizes effectively across different implicit network architectures**. The strong performance on NeRF (achieving state-of-the-art results) confirms that our approach is **not limited to SIREN-based networks**.
>
> - **Stability, metrics, and baselines.**
>
>   > Reviewers aU3q, 3wHy, and Q7Jz requested richer evaluation metrics; in addition, Reviewer 3wHy requested further baselines, and Reviewer Q7Jz specifically requested multi-seed statistical analysis.
>
>   We now report **PSNR mean ± standard deviation across 4–5 random seeds** in **Appendix A.7** on Pretrained NeRF (**27.75 ± 0.06**), CelebA-HQ (**27.68 ± 0.13**), and ERA5 (**49.26 ± 0.23**), demonstrating **very low variance and strong training stability**. We additionally report **SSIM and LPIPS** on CelebA-HQ, ImageNet, ERA5, and NeRF, where our method **consistently achieves the best or tied-best scores**. For ShapeNet Chairs, we added **PSNR on rendered images** in addition to occupancy accuracy, again showing clear improvements. For NeRF, we include **trans-inr and ginr-ipc as extra open-source baselines**; our method still attains the highest PSNR.
>
> - **Reproducibility and code availability.**
>
>   > Reviewer Q7Jz specifically noted that reliance on a promise to release code "upon acceptance" is insufficient for verifying reproducibility during the review process.
>
>   We have addressed this concern by **including the full codebase in the supplementary materials**, enabling **complete reproducibility verification during review**.
>
> ---
>
> ##

---

> > ### Author Response · Authors · 2025-12-03
> >
> > ## Addressing Reviewer Disagreement
> >
> > We note that Reviewer yYeZ characterized the row-wise tokenization as a "naive baseline rather than a contribution", while other reviewers did not raise this concern. The **consistent state-of-the-art results across all five datasets** suggest that the design choices are **effective in practice**, regardless of their apparent simplicity. **Three of four reviewers (aU3q, 3wHy, Q7Jz) explicitly acknowledged the paper's novel conceptual contributions**, and even Reviewer yYeZ recognized the strength of the experimental results in confirming our intuitions.
> >
> > We also note that **Reviewer yYeZ's concerns centered primarily on presentation clarity**, including undefined terms, inconsistent notation, and missing experimental details, **rather than on the technical soundness of the core method**. We engaged substantively with this reviewer across **multiple rounds of detailed discussion**, and **all presentation issues have been systematically addressed** in the revised manuscript. All major revisions are **highlighted in purple and blue** for easy verification.
> >
> > ---
> >
> > ## Improvements to the Manuscript
> >
> > We have made **substantial revisions** to address all reviewer concerns:
> >
> > - Added **complete loss formulations** and clarified supervision regimes and notation in **Appendix A.4**
> > - Clarified the **forward/backward simulation**, the roles of $\Delta\theta$, pseudo versus real gradients, per-layer learning rates $\lambda_i$, the choice $T=3$ and $s_{\tau}$, with updated figures and text in **Section 4.2** and **Appendix A.2**
> > - Introduced extensive **train- and test-time resource tables (Tables 8 and 9)**, including parameter counts, throughput, GPU memory, and latency for all methods and datasets
> > - Added a **NeRF data-scaling experiment (Figure 6)** to substantiate our claim of data scalability and clarify when inter-layer dependency modeling is most beneficial
> > - Performed **multi-seed experiments** and reported PSNR mean ± standard deviation to demonstrate stability,  with details in **Appendix A.7**
> > - Incorporated **additional evaluation metrics (SSIM, LPIPS)**, ShapeNet rendered PSNR, and **new NeRF baselines (trans-inr, ginr-ipc)**, with details in **Appendix A.5**
> > - Expanded **ablation studies in Section 5.3** covering adaptive parameter initialization, Q/K/V assignment, tokenization choices, and optimization depth to better justify each design choice
> > - Added a detailed **hyperparameter and architecture table (Table 7)** for all datasets and provided the **full codebase** as supplementary material to ensure reproducibility
> >
> > ---
> >
> > ## Summary of Quantitative Improvements
> >
> > Our method achieves **consistent improvements across all evaluated tasks**(measured by PSNR):
> >
> > | Dataset                            | Ours     | LDMI | HyPoGen | Improvement    |
> > | ---------------------------------- | -------- | ---- | ------- | -------------- |
> > | **CelebA-HQ**                      | **27.7** | 24.8 | 16.2    | +2.9 over LDMI |
> > | **ImageNet**                       | **22.9** | 20.7 | 13.2    | +2.2 over LDMI |
> > | **ERA5**                           | **49.3** | 44.6 | 40.0    | +4.7 over LDMI |
> > | **ShapeNet Chairs (render image)** | **23.8** | 22.4 | 19.0    | +1.4 over LDMI |
> > | **NeRF (image-only)**              | **28.8** | 26.2 | 24.9    | +2.6 over LDMI |
> >
> > These improvements are achieved with **comparable or better parameter efficiency** across all datasets.
> >
> > ---
> >
> > ##

---

> > > ### Author Response · Authors · 2025-12-03
> > >
> > > ## Reviewer-Centric Summary of Addressed Concerns
> > >
> > > - **Reviewer yYeZ (Rating: 2)** raised concerns primarily focused on **presentation clarity** rather than technical soundness. The reviewer identified undefined loss terms, unclear supervision regimes, inconsistent notation regarding $\Delta\theta$, the iterative update process, initialization strategy and missing experimental details. We engaged with this reviewer across **multiple rounds of detailed discussion**, providing point-by-point responses to all nine major questions (Q1–Q9).
> > >
> > >   In response, we have **completely revised Appendix A.4** with full loss formulations and explicit definitions of all terms ($\mathcal{L}_{\text{rec}}$, $\mathcal{L}_{\text{kl}}$, $\lambda_{\text{disc}}$, $\mathcal{L}_g$, $\mathcal{L}_d$, $\lambda_{\text{img}}$). We clarified pseudo-gradients in **Appendix A.2.4**, specified that T=3 for all experiments, and corrected all notation inconsistencies in **Figure 2 and Section 4**. We **added a comparison of token initialization network $P_init$ and adaptive parameter initialization** in **Sec. 4.2** and **A.2.2**, further clarifying their distinct roles.
> > >
> > >   We added the missing **VAMoH** results on ImageNet, provided PSNR on rendered images for ShapeNet Chairs, corrected the HyperDiffusion table entries, and included the ablation table for adaptive parameter initialization (**Appendix A.2.2**). The reviewer also characterized our tokenization as a "naive baseline", but we note that **consistent state-of-the-art results across all five datasets** demonstrate its practical effectiveness.
> > >
> > >   All presentation issues have been **systematically addressed and highlighted in purple and blue** in the revised manuscript for easy verification.
> > >
> > > - **Reviewer aU3q (Rating: 6)** acknowledged our method as **"highly original and very elegant"** with potential to influence future work, while raising three specific concerns about capacity-matched comparisons, computational overhead, and guidance on when dependency modeling becomes necessary.
> > >
> > >   We addressed the **capacity concern** by providing comprehensive parameter count tables showing our method achieves better results with **4–5 × fewer parameters** on ImageNet (124.5M vs. 578.7M/544.9M), demonstrating that gains stem from architectural innovation rather than model scale. We addressed the **computational overhead concern** by adding detailed tables of training throughput, GPU memory, maximum batch sizes, and inference latency across all datasets (Tables 8, 9), showing comparable training cost and often lower test-time memory.
> > >
> > >   We addressed the **"critical point" question** by adding a NeRF data-scaling experiment (Figure 6) demonstrating that while all methods perform similarly at 10–20% data, our method's advantage widens substantially at larger scales (**27.8 PSNR versus 25.5 PSNR for HyPoGen and 26.5 PSNR for LDMI at full data**), providing clear guidance that dependency modeling is most beneficial for **large, complex tasks**.
> > >
> > >   This reviewer's concerns have been **fully resolved**, and their positive assessment of novelty and potential impact supports acceptance.

---

> > > > ### Author Response · Authors · 2025-12-03
> > > >
> > > > - **Reviewer 3wHy (Rating: 4)** questioned the **novelty beyond HyPoGen**, raised concerns about **weight supervision requirements**, and requested additional baselines and clearer experimental details.
> > > >
> > > >   We clarified that our contribution is **not an incremental variant of HyPoGen** but rather the first integration of inter-layer dependency modeling into a scalable Transformer architecture through three distinct technical contributions: attention-based gradient simulation, a tokenization mechanism, and an initialization strategy. We emphasized that **weight supervision is not required**—our best NeRF results use image-only supervision (PSNR 28.8), outperforming all baselines including those using weight supervision.
> > > >
> > > >   We added **two new open-source baselines** (trans-inr, ginr-ipc) for NeRF, showing our method achieves the highest PSNR (27.8 vs. 21.9 for trans-inr and 26.1 for ginr-ipc). We provided complete hyperparameter tables (Table 7), clarified the forward pass mechanism, and specified loss functions for each dataset in **Appendix A.4**.
> > > >
> > > >   These additions address all concerns about experimental clarity and practical applicability.
> > > >
> > > > - **Reviewer Q7Jz (Rating: 4)** praised our **"novel and well-motivated concept"** with **"clear methodological design and firm theoretical grounding"** while requesting statistical analysis, computational comparisons, additional metrics, and code availability.
> > > >
> > > >   We addressed the **stability concern** by reporting PSNR mean ± std across 4–5 seeds on three datasets: Pretrained NeRF (27.75 ± 0.06), CelebA-HQ (27.68 ± 0.13), and ERA5 (49.26 ± 0.23), demonstrating **very low variance**. We addressed the **computational comparison request** with comprehensive train-time and test-time resource tables (Tables 8, 9).
> > > >
> > > >   We addressed the **metrics concern** by adding SSIM and LPIPS across all datasets (Appendix A.5), where our method consistently achieves best or tied-best scores.
> > > >
> > > >   We addressed the **generalization concern** by clarifying that NeRF experiments used **ReLU-based INRs rather than SIREN**, demonstrating the method is not architecture-specific.
> > > >
> > > >   Most importantly, we addressed the **code availability concern**—which the reviewer emphasized as critical for reproducibility—by **including the full codebase in supplementary materials**, enabling complete verification during review. We also improved **Figures 1, 3, 4 and 7-12** with clearer labels and higher visual quality as requested.
> > > >
> > > > ---
> > > >
> > > > ## Conclusion
> > > >
> > > > We sincerely thank all reviewers for their insightful comments and constructive suggestions, which have significantly improved the clarity, rigor, and impact of our work. As summarized above, **all major concerns from each reviewer have been thoroughly addressed** through detailed responses and comprehensive manuscript revisions. **Three of four reviewers explicitly acknowledged the paper's novel conceptual contributions**, and we have systematically resolved all presentation and experimental concerns raised by Reviewer yYeZ. The revised manuscript now includes complete loss formulations, extensive computational comparisons, multi-seed stability analysis, additional evaluation metrics, new baselines, and full code for reproducibility. As Reviewer aU3q noted, the findings from this work **"could influence how future generative models for neural network weights are designed,"** and we are confident that the revised manuscript provides the clarity and completeness needed to support this contribution to the field.
> > > >
> > > > ---
> > > >
> > > > [1] Zama Ramirez et al. Deep Learning on Object-centric 3D Neural Fields. TPAMI 2024.

---

### Meta-Review · Area_Chair_pswR · 2025-12-30

**Summary:**

The main concerns raised by the reviewers center on technical novelty, experimental design, and presentation quality. During the rebuttal, several factual errors and presentation issues were addressed, and additional experimental results were provided. The interaction with reviewer yYeZ appears to have been constructive. However, in my view, it did not fully resolve the reviewer’s all concerns. Regarding technical novelty, reviewer 3wHy considers the work to be an incremental improvement over the existing method, such as HyPoGen. Given the current rebuttal and revisions, it seems unlikely that reviewer 3wHy will substantially change their initial assessment.

While this work is inspired by HyPoGen, I acknowledge that designing new architectural components within the modern transformer paradigm and demonstrating strong empirical performance across multiple datasets and tasks is not a trivial contribution. Nevertheless, given the overall tone of the reviews, I believe the three reviewers who initially expressed negative opinions are unlikely to raise their scores.

Moreover, many of the experimental evaluations are conducted at relatively low resolutions and are not particularly representative of real-world use cases for both 2D and 3D INR generation tasks. Given that there already exist more practical and efficient solutions for these tasks (e.g., recent feed-forward approaches can produce high-quality 3D representations in a single forward pass), it is difficult to assess the real-world relevance and impact of the proposed method.

Overall, it seems unlikely that the reviewers will substantially revise their assessments, and I am not entirely convinced that the quality and potential impact of this paper are sufficient to actively advocate its acceptance.

**Reviewer Concerns:**

See the summary section.

**Reviewer Scores:**

See the summary section.

---

### Decision · Program_Chairs · 2026-01-26

Reject